# Integration of alternative fragmentation techniques into standard LC-MS workflows using a single deep learning model enhances proteome coverage

Nikita Levin [1,2,9], Cemil Can Saylan [3,9], Joel Lapin[3], Yana Demyanenko [1,2], Kevin L. Yang [4,5], John Sidda[1,2], Alexey I. Nesvizhskii [4,5], Mathias Wilhelm [3,6] ✉ & Shabaz Mohammed [1,7,8] ✉

Bottom-up proteomics relies predominantly on collision-induced dissociation (CID) for peptide sequencing, which has achieved remarkable sensitivity and efficiency now enabling single-cell analysis. However, CID shows limitations in characterizing post-translational modifications and complex proteoforms. Here we have developed an integrated mass spectrometry platform enabling automated collision-, electron- and photon-based fragmentation techniques. Using multi-enzyme deep proteomics workflows, we generated comprehensive datasets to train a unified Prosit deep learning model predicting spectra across all dissociation methods. This publicly available model, now integrated into FragPipe's MSBooster module, increased protein identifications by >10% on average for both data-dependent and data-independent acquisition across all fragmentation techniques. We demonstrate that alternative approaches, particularly electron-induced and ultraviolet photodissociation, which generate richer, more informative spectra, achieve identification efficiency competitive with CID while providing superior sequence coverage. This work establishes a framework enabling routine application of advanced fragmentation techniques in standard proteomics pipelines.

Mass spectrometry is the key approach for proteomics analysis[1] due to its specificity, sensitivity and speed. Sequence information is acquired by dissociating each peptide of proteolyzed proteins into fragments via an established process, recording the spectrum and identifying it using specialist software[1]. Collision-induced dissociation (CID) has endured as the gold standard for peptide sequencing on account of its fragmentation explainability, reproducibility and outstanding speed[2]. A variety of alternative dissociation techniques have been proposed as being complementary to CID. These include, but are not limited to, ultraviolet photodissociation (UVPD) using photons of various wavelengths[3–6] and irradiation by electrons in a wide range of energies, such as electron capture dissociation (ECD)[7–10], hot-ECD[8,10,11], electron transfer dissociation (ETD)[12–14] and electron ionization dissociation (EID)[15–17]. The electron-based methods, combined under the umbrella term 'ExD', differ from each other in the types of fragments generated. These techniques provide access to alternative fragmentation pathways that improve peptide sequence coverage and enable CID-labile modifications to be kept intact. Among them, ETD and electron transfer/high-energy collisional dissociation (EThcD)[18] were adopted in Tribrid Orbitrap instruments, advancing proteomics studies to a more accurate

analysis of phosphorylated[19,20] and glycosylated[21–25] peptides. However, all of the alternative approaches have had low uptake due to a catch-22 scenario: weak demand leads to less vendor attention on development and subsequent delivery of complicated, compromised, low-efficiency and low-speed designs. Given that these techniques remain specialized, the motivation to build suitable data analysis software lags behind classical CID approaches, which further exacerbates the issue.

In recent years there has been significant progress in the development of more accurate tools for analysis of bottom-up proteomics data[1]. In particular, the use of data-driven rescoring pipelines[26–31] based on predictions of peptide properties such as retention time or fragmentation spectra, has greatly increased the number of identifications compared with standard database searches[32,33]. Among deep learning-based tools, Prosit[27] and pDeep[31] have gained popularity for improving peptide identification through fragmentation spectrum prediction. The vast majority of these predictors are restricted to collisional data, while ExD methods and UVPD remain largely unsupported[32]. To date, only one model, PredFull[34], has demonstrated the ability to predict alternative ion types from ETD data; nevertheless, it still lacks sufficient training to robustly support UVPD and other ExD fragmentation methods such as ECD and EID. The development of tools for UVPD and ExD data analysis is hindered by the paucity of publicly available datasets for calibration of search algorithms and training of deep learning models. The generation of such data, in turn, requires specialized, often custom, mass spectrometry instrumentation. Recently, Papanastasiou et al. reported a novel instrument, the Omnitrap, which is a segmented ion trap enabling multiple ion-activation approaches in one mass spectrometry platform[35]. In collaboration, we built an Orbitrap-Omnitrap hybrid instrument that allows access to both common types of CID and multiple laser- and electron-based dissociation techniques, and demonstrated it for top-down proteomics[36]. Here, we further developed this instrument to operate on a liquid chromatography–mass spectrometry (LC-MS) timescale in an automated fashion. We set out to create tools for data analysis for these alternatives, to be on a par with current CID offerings and to make these dissociation techniques available for wider community adoption.

## Results

### Development of Omnitrap UVPD, ECD and EID LC-MS methods

The results of our recent development and characterization of UVPD, EID and ECD on the Omnitrap platform[36] suggested that it could be deployed in an LC-MS configuration for the analysis of complex peptide mixtures. Given that the conditions in direct-infusion experiments from our earlier work, such as number of available ions, injection times and ion transfer logistics, are typically more relaxed than in automated LC-MS analysis, an investigation is required to determine the optimal parameters for all dissociation techniques. Direct-infusion experiments reported previously[36] were focused on higher resolution and signal-to-noise ratio with no regard to duty cycle. Given that acquisition of spectra with this configuration has limited parallelization potential (Extended Data Fig. 1a), we initially concentrated on reducing scan length to increase speed of spectra acquisition to handle the complexity of proteomes. The Omnitrap design requires ions to be cooled through a gas pulse prior to any ion manipulation. The original design used a single gas valve that had a maximum repetition rate of 10 Hz (Extended Data Fig. 1b). To improve the maximum rate of the Omnitrap we implemented the use of two valves, operating alternately, for gas injection, which can potentially double the speed (Extended Data Fig. 1b). Subsequently, we optimized the potentials for ion transfer in the Omnitrap to reduce the background collisional fragmentation (Supplementary Notes and Extended Data Fig. 1c–f). We then focused on increasing the identification rate in LC-MS experiments through application of pragmatic parameters for acquisition (Fig. 1a). Unless otherwise specified,

human Expi293F cell lysate digests were used as the analyte. We began with the characterization of UVPD. We first varied the number of laser pulses at a fixed energy of 3 mJ per pulse and then varied the energy for a fixed number of pulses. For data analysis, we started with using only $b$ and $y$ ions for identification, which were previously shown to be the most abundant in UVPD of tryptic peptides[6,37,38]. Analysis shows that increasing the number of laser pulses leads to a greater number of identified peptide–spectrum matches (PSMs) and peptide sequences until a maximum is reached at four pulses (Fig. 1b). Further increases in the number of laser pulses used for dissociation results in a drop of the identification rate, either due to secondary fragmentation or reduced scan rate. We selected four pulses for further investigation and varied the energy of each pulse. In this series of experiments, the maximum of identified PSMs and peptide sequences was observed at distinct energies depending on the type of fragment ions used for identification (Fig. 1c). Using only $b$ and $y$ fragments, the maximum is observed at 5 mJ per pulse, while when other types of fragment characteristic of UVPD are used, namely $a, c, x, z$ (ref. 4) (see Supplementary Table 1 for structures and definitions of fragment ions considered in this work), the maximum is located at 6 mJ per pulse. Given that $a, c, x, z$ in contrast to $b, y$ are more unique to UVPD, we opted to use 6 mJ per pulse in future experiments.

Next, we studied the optimal reaction times for ExD. In typical ExD experiments, ions are transferred into the reaction chamber and undergo irradiation by electrons emitted by a heated filament[35] during a specified amount of time (Extended Data Fig. 1g,h). In EID experiments, we varied the irradiation time from 25 ms to 150 ms and measured the number of identified PSMs and peptides. We observed that $b$ and $y$ ions can be the most prominent ions in EID. When using only these two ions for analysis, the number of PSMs and of peptides reaches the maximum value at 50 ms of irradiation (Fig. 1d). At longer irradiation times, these numbers start to drop. Interestingly, the profile of peptide identification shows a much more distinctive dependence on the type of ions used for analysis compared with UVPD (Fig. 1d). At shorter irradiation times, $a, c, x, z$ fragments are underrepresented compared with those of $b, y$, and the largest number of PSMs and peptides was observed at 75 ms (Fig. 1d). To keep scan rates high in the interest of absolute number of identifications, we chose to continue with the 50 ms irradiation time. Finally, we found 50 ms of irradiation to be optimal in ECD using $c$ and $z$ fragments for the data analysis (Fig. 1e). We did not investigate other main-series types of fragments, because the majority of the products of ECD of relatively short peptides are $c$ and $z$ ions[7,8]. Given that ECD is known to be a charge-dependent process favoring higher charge states, the value of 50 ms obtained using mainly doubly charged and less frequently triply charged precursors of tryptic peptides can be considered conservative. To characterize the fragmentation behavior of ECD, UVPD and EID, a larger and more diverse range of peptides is required.

### Large-scale multi-enzyme LC-MS analysis

We increased the diversity of peptide sequences through the use of more proteases, and we increased peptide depth by utilizing offline reverse-phase high-pH fractionation (Fig. 2a). We chose trypsin, LysC, GluC, chymotrypsin and LysN because they have been shown to produce complementary results in terms of peptide length, protein sequence coverage, and frequencies and positions of amino acid residues across the peptide backbone[39]. Next, we fractionated each digest[40] into 20 pooled fractions and analyzed all of them using ECD, EID, beam type CID (referred to as higher-energy CID or 'HCD' on Thermo instrumentation) and UVPD LC-MS. The choice of liquid chromatography gradient time for the dissociation techniques was based on their maximum sequencing rate to ensure that they all produced a similar number of scans.

The analysis of UVPD, EID and ECD data is not as straightforward as that of HCD data. The major products of HCD are well characterized,

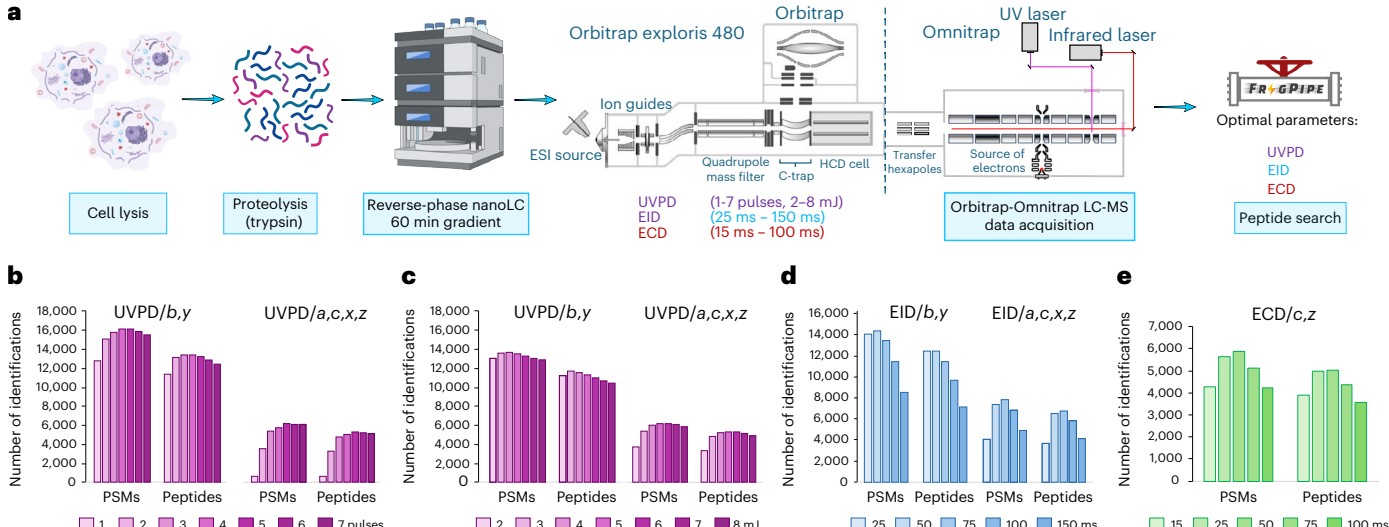

**Fig. 1 | Optimization of ECD, EID, and UVPD parameters in bottom-up experiments. a**, Experimental workflow. **b–e**, Number of PSMs and peptides identified in UVPD experiments varying the number of UV laser pulses at 3 mJ pulse$^{-1}$ (**b**), UVPD experiments using four laser pulses and varying the pulse energy (**c**), EID experiments varying the irradiation time at 25 eV of electron energy (**d**), and ECD experiments varying the irradiation time at -1 eV of electron energy (**e**). In UVPD and EID, $b$, $y$ or $a$, $c$, $x$, $z$ fragments were used for data analysis; $c$ and $z$ ions were used in the analysis of ECD data. Schematic diagram in **a** created in BioRender; Govender Kirkpatrick, M. https://biorender.com/qqloq0m (2025).

with $a$, $b$, $y$ ions dominating data. In contrast, UVPD and EID are known to produce all main-series types of peptide fragments as well as some radical $a + 1$, $x + 1$ ions,[4,15,41] with the last two largely understudied. The average proportion of each type of main-series fragment has been reported for UVPD[6,37,38]; however, the effects of using these ions and their combinations in the automated data analysis have not been extensively discussed. We therefore analyzed the acquired raw data using several unique combinations of the expected fragment types with the goal to maximize the number of identified PSMs while maintaining the same 1% false discovery rate (FDR). For ECD, the most important ions for robust identification were $c$ and $z$ (Fig. 2b). The addition of $c - 1$ or $z + 1$ had a minimal and slightly detrimental effect. Analogously, $b$ and $y$ were the dominant ion types for both EID and UVPD. However, $a$, $a + 1$, $c$, $z$ ions were beneficial for improving identification rates for EID, while $b$, $y$ produced the best results in UVPD. The numbers when broken down to the individual enzyme level are similar to the global result, although tryptic and LysC peptides enhance the formation of $z + 1$ ions while impairing the formation of $c - 1$ in ECD, and favor the generation of $y$ ions in EID and UVPD compared with other enzymes (Supplementary Fig. S1). The results for UVPD and EID seem to be strongly dependent on $y$ ions and to a smaller degree on $b$ ions. While no extensive literature exists for EID, our UVPD data agree with previous findings. Others also found that $b$, $y$ fragments are the most abundant types of ions in 193 nm UVPD of tryptic peptides, and the ion current of $y$ fragments is approximately double that of $b$ (refs. 6,37). Similarly, $b$, $y$ fragments dominate the spectra in 213 nm UVPD of tryptic peptides, and the average number of annotated $y$ fragments is twice that of $b$ ions[38].

In total, each fragmentation technique produced between approximately 3.5 million and 4.5 million MS2 spectra across five enzymes, 20 fractions per enzyme (Fig. 2c). EID data had the least number of PSMs (~900,000), while UVPD, which has the fastest acquisition rate among all Omnitrap techniques studied here (~6.3 MS2 scans per second on average), had 1,141,000 (Fig. 2c). Surprisingly, charge-dependent ECD came closest to UVPD with 1,070,000 PSMs, even though its scan rate (~5.2 MS2 spectra per second) was essentially the same as in EID. HCD showed the highest numbers with 1,160,000 PSMs acquired using 60 minute gradients at the rate of, on average, ~13 MS2 scans per second. Pleasingly, the efficiency of peptide sequencing by EID (24.8%) and UVPD (25.6%), expressed

as the ratio of the number of confidently identified PSMs to that of acquired MS2 scans, is essentially the same as by HCD (24.9%), while the efficiency of sequencing by ECD (30.3%) was the best (Fig. 2c). This was surprising considering the relative inefficiency of ECD for doubly charged peptides, which represent a substantial subset of identified peptides (Extended Data Fig. 2a).

The MSFragger hyperscore can serve as an indirect measure of the number of fragments found in a spectrum, similar to a spectrum quality score[42]. We plotted density contour plots for hyperscores of all unique precursors (that is, unique combinations of amino acid sequences, charge states and modifications, Extended Data Fig. 2b,c) per charge state using $c$, $z$ fragments in ECD and $b$, $y$ fragments in UVPD, EID and HCD (Fig. 2d and Supplementary Figs. S2 and S3). Expectedly, the distribution of hyperscores in ECD is strongly charge dependent, with doubly charged precursors assigned substantially lower values. Furthermore, the hyperscore distributions for 3+ and 4+ precursors in ECD have an apparent maximum at 800 Th. A similar trend was reported earlier by Good et al. for ETD of tryptic and LysC peptides, in which the percent of bonds cleaved by ETD begins to drop at approximately 600 Th for 3+ precursors and 650 Th for 4+ ones[13]. When analyzing solely $b$, $y$ ion series, EID, UVPD and HCD all produce very similar hyperscore distributions for the same charge states of precursors (Fig. 2d). UVPD has marginally higher hyperscores in the low-$m/z$ range than HCD, and EID produces lower hyperscores in the high-$m/z$ range than UVPD and HCD. The upper boundary of hyperscore distributions for these dissociation techniques starts to drop beyond approximately 2,000–2,500 Da for 2+ and 3+ precursors and 2,500–3,000 Da for 4+ precursors. We interpret these observations as the reduction of the signal-to-noise ratio that follows the spreading of available fragment signal across a larger number of produced fragments in spectra of long and highly charged peptides, that is, signal splitting. The difference in number of identifications with the same 1% FDR was marginal for UVPD and EID when we increased the number of fragment types all the way up to $a$, $b$, $c$, $x$, $y$, $z$, as long as the $b$, $y$ fragments were included (Fig. 2b). We therefore investigated how the choice of type of fragment for analysis affects hyperscores (Fig. 2e and Supplementary Fig. S4). Clearly, adding more types of fragments results in greatly improved hyperscores for both EID and UVPD, indicating a larger number of dissociated bonds and data-rich spectra.

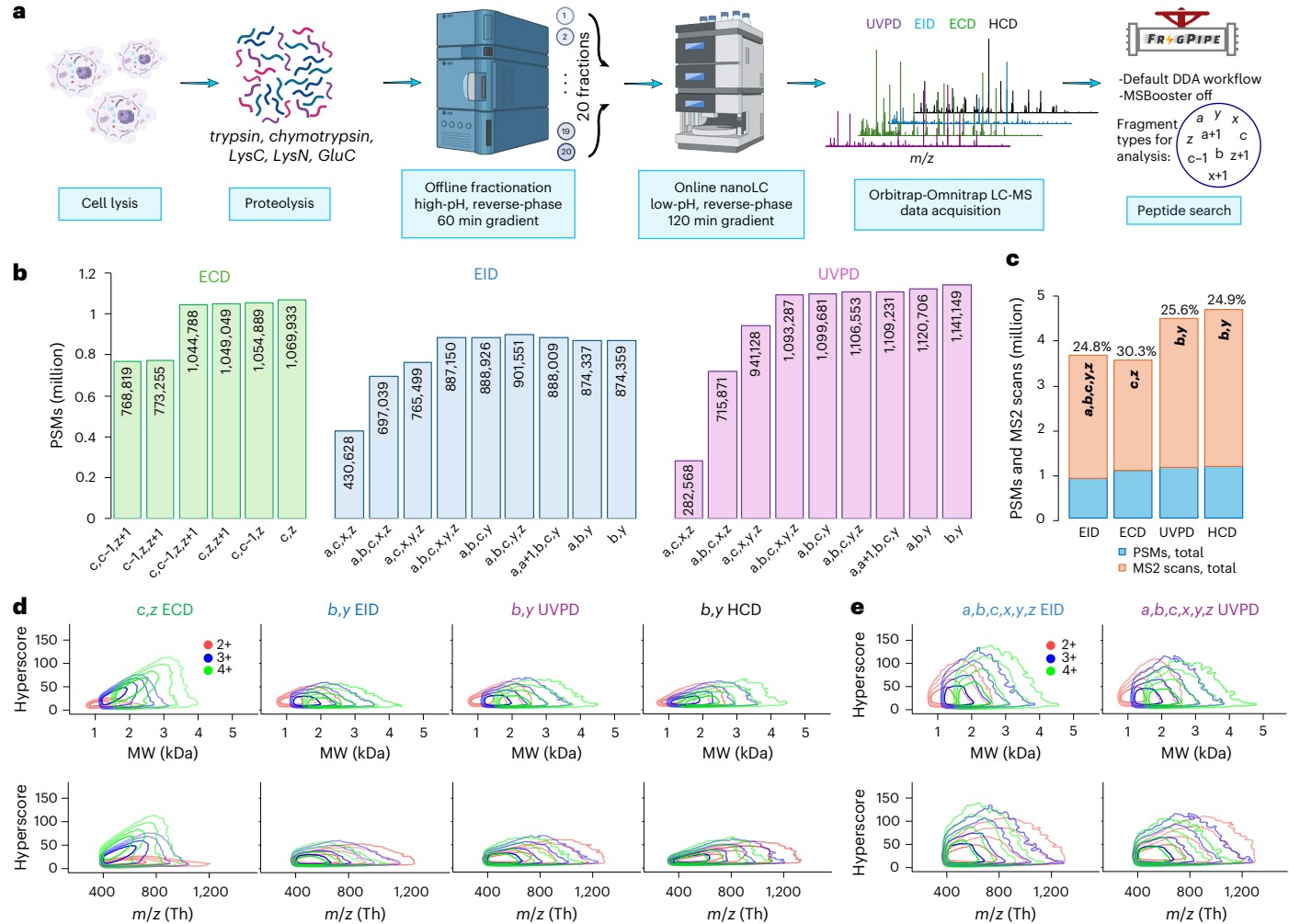

**Fig. 2 | Large-scale bottom-up ECD, EID, UVPD, and HCD analysis.**
**a**, Experimental workflow. **b**, Total numbers of PSMs in ECD, EID, and UVPD experiments identified using different combinations of fragment types. **c**, Highest number of PSMs from **b** (blue) and total number of acquired MS2 scans (orange) in ECD, EID, UVPD and HCD experiments; the rate of PSM identification is shown above each corresponding bar. **d**, Density contour plots of hyperscore distributions of 2+, 3+ and 4+ charge states of unique PSMs (unique combination of amino acid sequence, charge and modification selected by highest hyperscore) acquired in ECD using $c$ and $z$ fragments and in EID, UVPD and HCD using $b$ and $y$ fragments. **e**, Density contour plots of hyperscore distributions of 2+, 3+ and 4+ charge states of unique PSMs acquired in EID and UVPD using $a,b,c,x,y,z$ fragments. Contour lines demarcate the smallest regions to contain 50%, 80%, 95% and 99% of points. Schematic diagram in **a** created in BioRender; Govender Kirkpatrick, M. https://biorender.com/4anifnk (2025).

## Deep learning modeling of UVPD, EID and ECD fragment intensities

PSM scoring can be improved substantially if performed against experimental or in silico-generated spectral libraries[32]. Deep learning models have demonstrated promising results in predicting CID-based spectra of peptides using only peptide sequence, charge state and collision energy as input[26–28,31], but no such models exist for other fragmentation techniques due to the lack of large amounts of high-quality data for training. We therefore set out to use the datasets generated in this work to train a deep learning model able to predict fragment ion intensities. To create a more comprehensive model we then generated a similar dataset for electron-transfer/collision-induced dissociation (ETciD) on a Thermo Tribrid instrument (Supplementary Notes). Training a deep model requires converting the raw data into a dataset containing correctly annotated peak intensities. This implies that we need to solve potential clashes such as, for example, $a + 1$ ion, which is a radical $a$ ion coupled with an additional hydrogen atom, versus the $^{13}$C peak for an $a$ ion. For all datasets, we performed an automated annotation of major fragment types expected in EID, ECD, ETciD and UVPD (Supplementary Table 1) using the Oktoberfest framework[30].

The comparison of $[a + 1]/[a]$ ratio in HCD, EID and UVPD suggests that a large proportion of $a + 1$ in EID and UVPD spectra originate from gas-phase electron- and photon-based chemistries (Fig. 3a, Extended Data Fig. 3, Supplementary Figs. S5–S9 and Supplementary Notes). With the annotated spectra in hand, we defined our model's ion dictionary and curated training and validation datasets. The original Prosit model[27] architecture was designed around a structured output space consisting of $b$ and $y$ fragments with lengths 1–29 and charges +1 to +3. By contrast, the model trained on our data has an unstructured output space, with fragment ions chosen based on frequency of occurrence (≥100 occurrences, Supplementary Figs. S5–S9). The model also takes the categorical fragmentation type as input; given that the HCD data were acquired on a single instrument, it was unnecessary to use collision energy as additional input to the model, as was performed for previous Prosit models[27]. Our model shares similarity with the original Prosit model in that the sequence and metadata are separately encoded into latent spaces and combined in the interior of the network, but the metadata have slightly changed, and the model outputs predicted intensities of 815 fragment ions of various length, charge and fragment type (Fig. 3b). Results show very little overtraining: the median Pearson

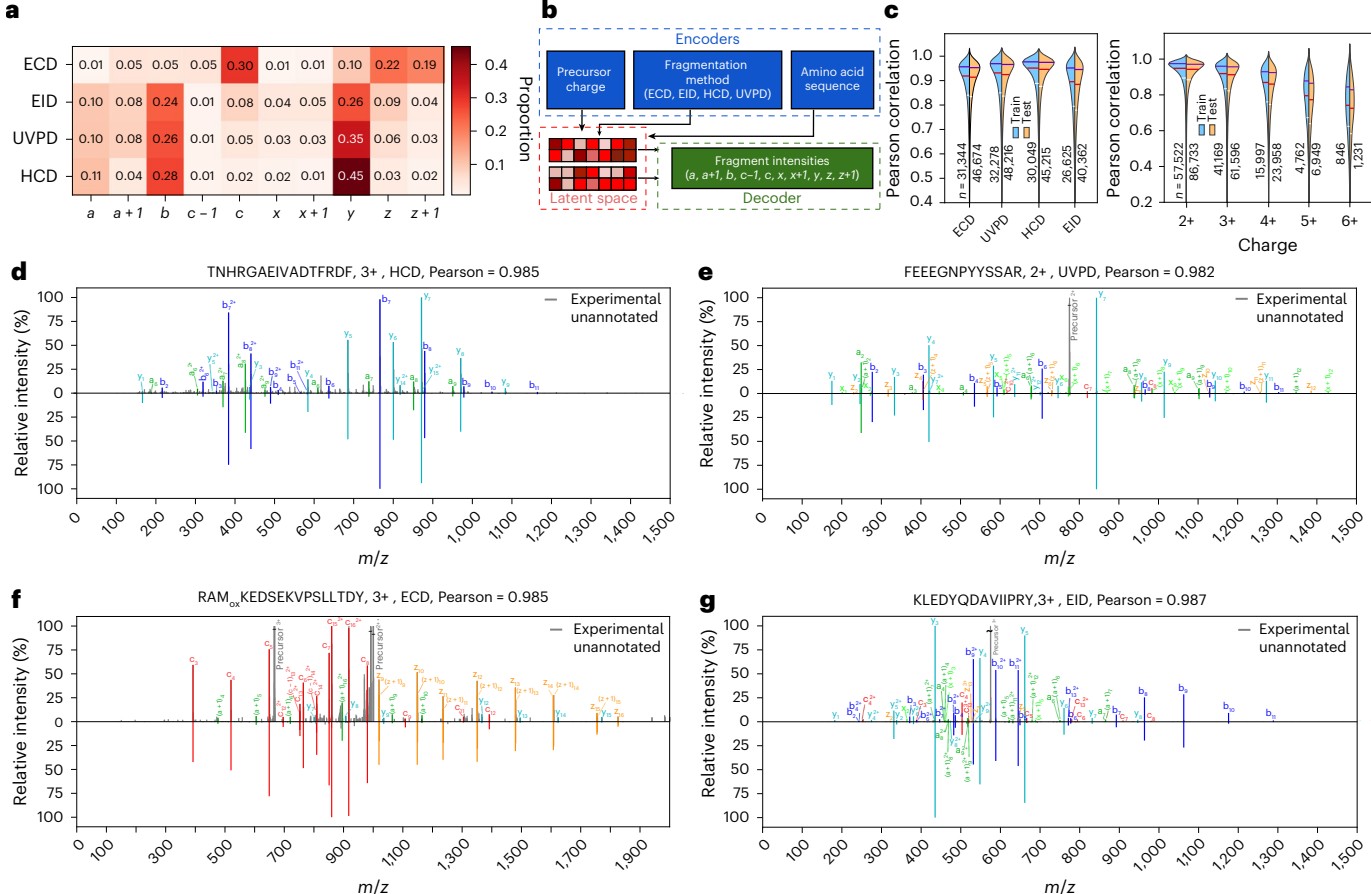

**Fig. 3 | Deep learning training pipeline, from annotation to evaluation.**
**a**, Heatmap of mean proportion of each type of fragment ion among all
annotated peaks in ECD, EID, HCD and UVPD spectra acquired across all enzymes,
not reflecting relative intensity of ions. Annotation was performed for 10 ion
types: $a, a+1, b, c-1, c, x, x+1, y, z, z+1$ (Supplementary Table 1). **b**, The modified
Prosit deep learning architecture for prediction of fragment ion intensities in
ECD, EID, HCD and UVPD spectra. The input parameters (peptide sequences,
precursor charge state and fragmentation method) are encoded into a latent
representation (latent space). This representation is then decoded to predict
fragment ion intensities. **c**, Pearson correlation coefficients between predicted
and experimental spectra in training and test sets separated by fragmentation
method (left) and charge state (right). Horizontal white, red, and blue lines
correspond to 25%, 50% and 75% percentiles, respectively. n indicates sample
size. Distributions extending beyond 1.0 are plotting artefacts. **d**–**g**, Mirror plots
of selected precursors in HCD (**d**), UVPD (**e**), ECD (**f**) and EID (**g**) data. Each mirror
plot compares experimental (top) and predicted (bottom) fragment intensities,
with each fragment type uniquely colored.

correlations for ECD, UVPD, HCD and EID are 0.919, 0.931, 0.950 and
0.897, respectively, on the training set, and the corresponding scores
for the test set are only ~0.005 lower for each fragmentation method
(Fig. 3c and Extended Data Fig. 4). Furthermore, we observe that precur-
sor charge is consequential for prediction performance, with precursor
charges greater than 2 having an increasingly wide range of Pearson
correlations, likely to be due to the sparsity of high charge precursors
in the training set and increasingly complex fragment ions present in
the spectra. Pleasingly, we see that conditioned on the fragmentation
method the model reliably assigns appreciable intensity only to those
fragments expected for each fragmentation method, for example $b$,
$y$ for HCD and $c$, $z$ for ECD (Fig. 3d,e). The model is also able to predict
intensities of $b$, $y$ and minor fragments, such as $a$, $a+1$, $x$, $x+1$, $c$, $z$ in
UVPD and EID, although predictions of low-intensity ions for the latter
seem slightly less accurate (Fig. 3f,g). We performed a series of addi-
tional tests to validate the robustness and correctness of our model
(Supplementary Notes and Supplementary Fig. S10).

## Rescoring of alternative fragmentation data using fragment intensity predictions

An efficient control of FDR in database searching is critical for iden-
tification of true-positive peptide matches. Previously, we showed

that data-driven rescoring of CID data using the Prosit model greatly
improved number and accuracy of peptide identifications[27]. We
hypothesized that predicting fragment ion intensity would be ben-
eficial for improving the results of the database searches of UVPD,
EID and ECD data as well. Using the optimized MSFragger results we
first calculated the ratio of the number of all observed to that of all
possible theoretical fragment ions in each identified spectrum (Fig. 4a
and Extended Data Fig. 5, upper distributions). The resulting distribu-
tions for target and decoy (a priori false-positive) PSMs were heavily
intermixed and shifted towards smaller ratios. EID and UVPD ratios
were particularly small due to a large number of theoretical ions. We
then calculated the same ratios but allowed only fragments predicted
by Prosit (Fig. 4a and Extended Data Fig. 5, lower distributions). The
inclusion of only predicted fragments split the distribution of ratios
of target PSMs, in which the majority shifted towards higher values
with a larger portion being above 0.8, and the remainder were essen-
tially unchanged. At the same time, the ratio of decoy PSMs remained
clustered at lower values. This indicates a substantial improvement
in the alignment between the observed and predicted fragment ions.
Next, we applied data-driven rescoring using the Oktoberfest
framework, which benefits from the here-developed fragment ion inten-
sity prediction model by generating fragment intensity-dependent

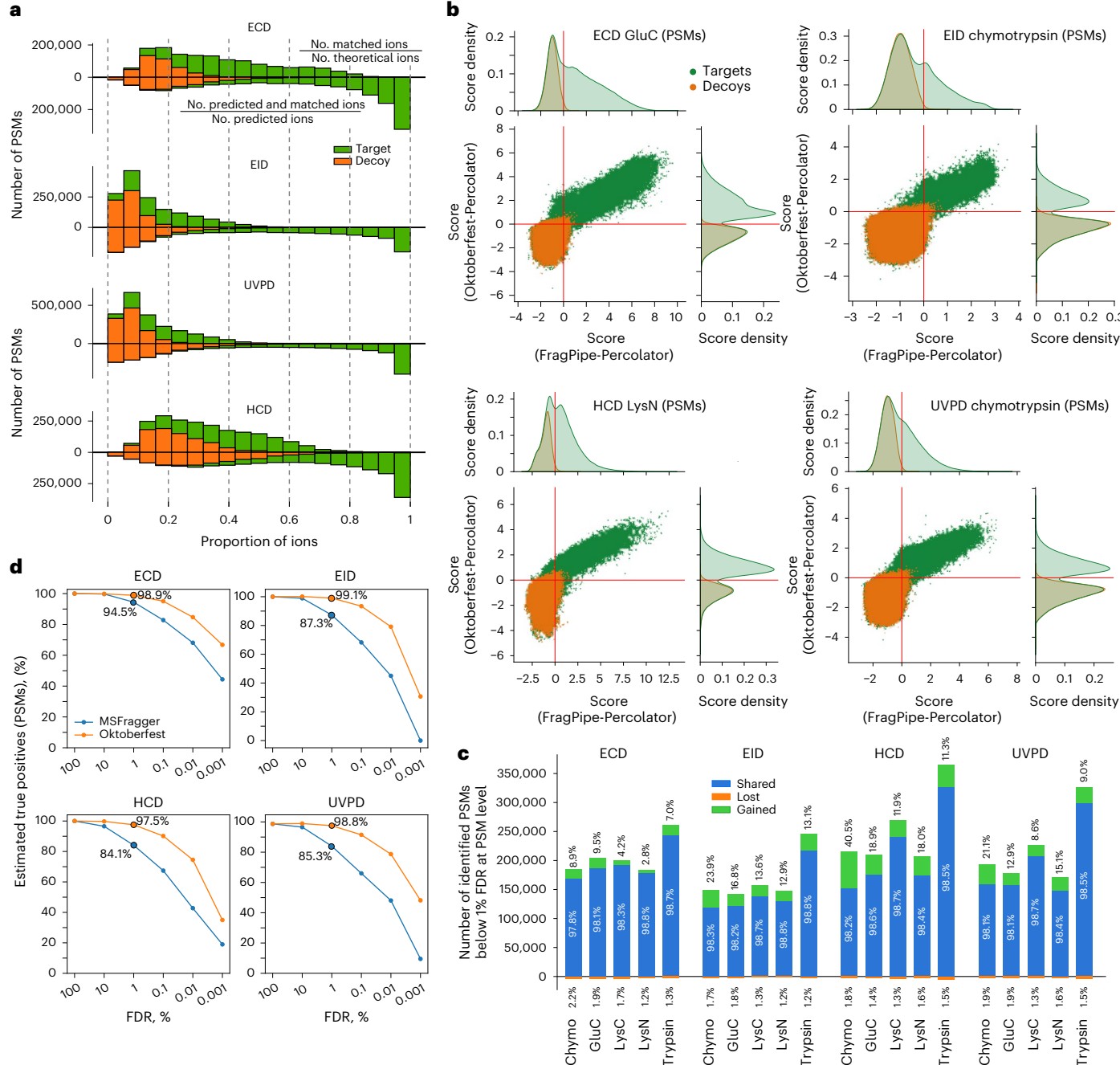

**Fig. 4 | Intensity prediction improves database search quality of ECD, EID, HCD and UVPD data. a**, Histogram of the ratio of experimentally observed ions to all theoretically possible fragments (upper distributions); and histogram of the ratio of predicted and experimentally observed ions to all predicted ions (lower distributions). **b**, Correlation of Percolator scores for all target and decoy PSMs obtained from the rescoring of the MSFragger (top) and Oktoberfest (right) sets of scores for selected combinations of enzyme and fragmentation technique. The red solid lines indicate the 1% PSM-level FDR cut-offs. For database search scores, the best combinations of fragment types from Fig. 2b were used; for Oktoberfest scoring, most frequently annotated fragment types

(>4% of all annotated ions across all spectra) were used for each dissociation method (Extended Data Fig. 3). **c**, Number of shared, gained and lost PSMs identified at 1% PSM-level FDR using the Oktoberfest set of scores compared to the original MSFragger search for each fragmentation technique per enzyme. The numbers correspond to the data from **b** and Supplementary Figs. S11–S14. Chymo, chymotrypsin. **d**, Proportion of the number of true-positive PSMs to the estimated maximum number of true-positive PSMs acquired using original MSFragger and Oktoberfest scores at different values of PSM-level FDR for each fragmentation technique, all enzymes combined.

scores rather than relying only on the presence or absence of any theoretical fragments. In combination with Percolator[43], these scores are aggregated into a single score that maximizes the separation of correct and incorrect matches. The resulting Oktoberfest scores were then compared to the Percolator-derived scores from MSFragger (Fig. 4b and Supplementary Figs. S11–S15), which do not include fragment

intensity-based features. For MSFragger database searches, we chose the best combination of ion types for each fragmentation method from Figure 2b, and for rescoring in Oktoberfest we used all of the most frequently annotated types of fragments (>4% of annotated ions in a spectrum, averaged across all spectra) for each fragmentation technique (Extended Data Fig. 3). Both sets of scores were filtered to

1% FDR using Percolator[43]. While rescoring led to remarkable separation of decoys from targets for the majority of enzyme–fragmentation method pairs (Fig. 4b and Supplementary Figs. S11–S15), ECD in general demonstrated sufficient separation in database searches, such that rescoring delivers only marginal improvements in identification (Supplementary Fig. S11). This partly explains the highest identification rate observed for ECD in the initial database searches (Fig. 2c). We attribute this to the relative cleanliness of ECD spectra that consist primarily of *c*, *z* fragments, precursor ions and charge-reduced species, thus reducing chances for random false matches. Interestingly, ECD was the only technique in which it was possible to discriminate the distributions of charge states among target PSMs after rescoring, which reflects the distinct charge-dependent kinetics of this process (Supplementary Fig. S16). Using rescoring, we were able to salvage a substantial number of PSMs in all combinations of enzyme and dissociation method (quadrant II in Fig. 4b and Supplementary Figs. S11–S15). At the same time, a high number of PSMs initially identified were discarded (quadrant IV in Fig. 4b and Supplementary Figs. S11–S15).

To evaluate how this separation of scores translated into gains and losses of PSMs and peptides, we compared the results of the database search and rescoring at both 1% PSM-level (Fig. 4c) and 1% peptide-level FDR (Supplementary Figs. S17 and S18). The number of gained PSMs varied (depending on the enzyme and fragmentation method) between approximately 3% and 40.5%, with chymotrypsin HCD data producing a notable gain of 40.5%. The latter observation is consistent with our previous findings[27]. Remarkably, chymotrypsin was also the main beneficiary of rescoring in UVPD and EID data. This demonstrates the usefulness of rescoring for expanded search spaces characterized by an increased number of possible charge states, allowed missed cleavages and reduced enzyme specificity, all of which are typical for chymotrypsin (Extended Data Fig. 2a). Consistent with the score distributions (Fig. 4b and Supplementary Figs. S11–S15), ECD had the lowest number of gained PSMs and peptides regardless of protease among all fragmentation techniques (Fig. 4c and Supplementary Fig. S17). Further investigation of ECD data shows that prediction of retention time and of fragment intensity generated similar gains, each adding approximately 6.5% of PSMs (Supplementary Notes and Extended Data Fig. 6). Such a relatively modest contribution of retention time predictions shows that improvements observed after rescoring of other combinations of enzyme and fragmentation technique are primarily driven by the new Prosit model.

To explore the reasons for the varying number of gains observed, we investigated the recovery of estimated true-positive PSMs. We compared the number of estimated true positives across a range of FDR thresholds (by subtracting the number of decoy PSMs from the number of target PSMs at different FDR cut-offs) before and after rescoring with the total number of estimated true positives in the dataset that could be recovered from the initial search results, by subtracting the total number of decoys from the total number of target PSMs (Fig. 4d and Supplementary Fig. S19). At 1% PSM-level FDR, rescored ECD, EID and UVPD searches recovered more than 97% of possible true positives, while the original database searches extracted approximately 95% in ECD, 87% in EID, 85% in UVPD, and 84% in HCD. At a stricter FDR of 0.01%, the results after rescoring still captured more than 75% of all estimated possible true positives, with ECD showing the highest proportion approaching 85%. At the same FDR level, initial database searches identified less than 70% of possible true positives in ECD and less than 55% in all other dissociation methods (Fig. 4d). The analysis shows that data-driven rescoring using the pan-fragmentation Prosit model substantially increases the proportion of estimated true-positive PSMs retained at stringent thresholds, approaching saturation of the set of PSMs recoverable from the initial MSFragger search results. It is important to note that further correct identifications, for example from modified peptides not considered in the initial search, cannot be considered in the estimation of the number of true positives.

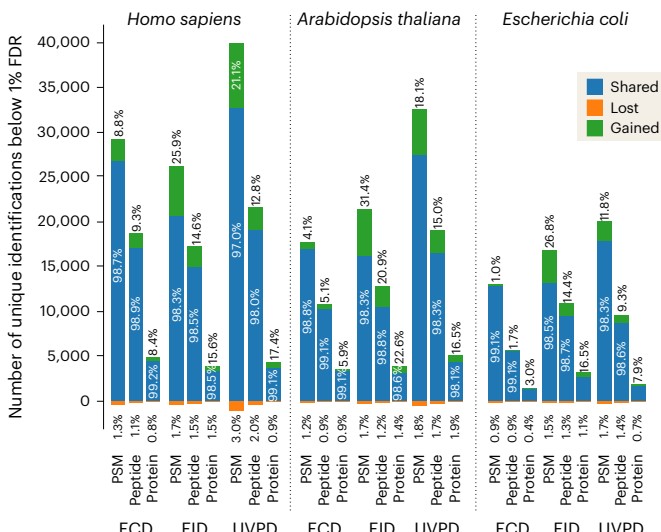

**Fig. 5 | Intensity prediction improves search quality of ECD, EID and UVPD DIA data.** Number of PSMs, peptides and proteins identified at 1% FDR in the UVPD, EID and ECD DIA data of unfractionated tryptic digests of human, *A. thaliana* and *E. coli* proteins. The analysis was performed in the FragPipe platform using the MSFragger search engine with Prosit predictions of fragment ion intensities implemented within the MSBooster module. The numbers of shared, gained and lost identifications correspond to the analysis with MSBooster 'on' as compared with the results obtained with MSBooster 'off'.

The rescoring data provided an opportunity to inspect the efficacy of each enzyme and dissociation technique for proteome analysis (Supplementary Notes, Extended Data Figs. 7 and 8 and Supplementary Figs. S20–S24). Trypsin, as expected, identified the most PSMs, peptides and proteins for every fragmentation technique. Chymotrypsin had the next best result, with LysC and LysN slightly further behind (Extended Data Fig. 7a and Supplementary Fig. S20a), replicating previous trends observed for CID and ETciD data[44–46]. The enzyme GluC clustered with LysN, appearing to be slightly superior or inferior depending on the dissociation technique. Average protein sequence coverage was similar for each fragmentation technique (Extended Data Fig. 8). To assess complementarity at the protein sequence level we represented our data at the amino acid level. In general terms, when comparing the complementarity of trypsin against its alternatives, we saw substantial improvements in proteome coverage for all fragmentation techniques (Extended Data Fig. 7b and Supplementary Fig. S20b); in fact, the unique combined coverage for LysN, LysC, GluC and chymotrypsin was more than that for trypsin. These observations echo previous work demonstrating the complementarity of enzymes for improving sequence coverage[39,44–46]. It should be noted that each trypsin fraction was essentially analyzed with LC-MS four times, and a more exhaustive LC-MS analysis would not significantly increase proteome coverage, and hence the amount of analysis time for the other enzymes versus trypsin is not an important factor in the comparison. Further analysis of unique coverage for each fragmentation technique showed that UVPD produced the most amount of unique data, with HCD and ECD close behind, and EID the least (Extended Data Fig. 7c). However, UVPD had significant overlap with EID, which might be a reason for the weak unique proteome coverage result for EID (Extended Data Fig. 7c).

## Application of data-independent acquisition in all fragmentation techniques
The spectral prediction model created in this work is portable and freely available as 'Prosit_2025_intensity_MultiFrag' at the Koina model repository[47], and can be interfaced from within any software suite.

We implemented our model within FragPipe as part of MSBooster[29]. We reanalyzed the deep proteome data in MSFragger to compare the results with and without MSBooster and found very similar gains to those observed using Oktoberfest at both the PSM and peptide levels (Extended Data Fig. 9). Combined with the optimization of search parameters in FragPipe, we can now perform both data-dependent and data-independent acquisition (DDA and DIA, respectively) analyses (pseudo-DDA through the use of DIA-Umpire) for all activation techniques. The ability to now utilize these activation techniques with DIA approaches led us to create DIA methodologies for the Orbitrap-Omnitrap. The change in ion population, both in terms of ion density and distribution of charge states, required adjustment of the acquisition parameters for each dissociation technique both at the Exploris and Omnitrap level (see Methods). We carried out LC-MS analyses on unfractionated tryptic cell lysate digests from *Homo sapiens* (Expi293F), *Arabidopsis thaliana* and *Escherichia coli* cells. We introduced the last two types of cells to assess the universality of the Prosit model. To optimize duty cycle, we chose to use the 'normal isolation window' approach with MS1 range bound to retention time[48]. MSBooster, using the here-developed Prosit model, increased identification rate at the PSM, peptide and protein levels for all three cell types. The *A. thaliana* and *H. sapiens* lysate samples had the largest improvements, trading top position depending on exact context. On average, ECD had the lowest gains across all samples, with the worst result being 1.0%, 1.7% and 3.0% at the three levels for *E. coli*, while EID demonstrated the largest improvements across all three types of samples, with the best result being 31.4%, 20.9% and 22.6% at the three levels for the *A. thaliana* sample (Fig. 5).

## Discussion

The intrinsic challenges of proteome characterization have necessitated a focus on sensitivity and speed to reach biologically useful levels of proteome information. In the last few years, the field has been able to reach a feasible level of proteome coverage in minutes and on sample sizes as low as a single cell[49,50]. However, the great variety of protein and peptide physicochemical properties embedded in the proteome places a hard limit on what modern proteomics approaches based on trypsin and CID can potentially observe. The blind spots of these approaches are well described, and alternatives such as complementary proteases and activation techniques have been introduced although not widely adopted. In this work, we have demonstrated that UVPD, EID and ECD can generate equivalent levels of data to that of HCD from the same number of spectra, that is, these approaches can reach similar levels of efficiency on a proteome scale across the most popular enzymes. These data enabled us to train a model that can predict spectra approximately as accurately as those for CID. However, it should be noted that this model is not complete. These fragmentation techniques, in a similar fashion to CID, possess multiple parameters that will create subtly different instrument-specific spectra (Supplementary Notes, Supplementary Fig. S25 and Extended Data Fig. 10). Additional refinement of the model through transfer learning will be required for each instrument type. Nevertheless, the advent of spectra prediction will now enable the community to consider whether alternative dissociation techniques are more appropriate for their protein characterization needs. An advantage of EID and UVPD is that their spectra are richer and less affected by the properties of the sequence to be studied, and will certainly make more challenging proteoforms more accessible for characterization. We have demonstrated that our model could be used for the improvement of database search results both from DDA and DIA sources, creating software parity with cutting edge modern proteomics pipelines and providing excellent starting points for transfer-learning and fine-tuning approaches to extend the model to, for example, modified peptides or enhance the model to mitigate differences in instrumentation and acquisition methods[28,51–53]. The Orbitrap-Omnitrap configuration as described here is a potent characterization instrument that is arguably the most capable for the widest range of protein types, but it is not optimized for sensitivity at speed. We substantially reinforce the earlier observations[4,9,54] that the characteristics of each fragmentation method have benefits. We hope that increasing the ability of the community to observe benefits of these activation approaches will increase demand and encourage vendors to consider implementation of more efficient designs, to enable most, if not all, application types to be as successful on these activation platforms as the standard CID experiment, with the added advantage of far richer data and more confident characterization.

## Online content

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

[1]Rosalind Franklin Institute, Harwell Campus, Didcot, UK. [2]Department of Pharmacology, University of Oxford, Oxford, UK. [3]Computational Mass Spectrometry, Technical University of Munich, Freising, Germany. [4]Gilbert S. Omenn Department of Computational Medicine and Bioinformatics, University of Michigan, Ann Arbor, MI, USA. [5]Department of Pathology, University of Michigan, Ann Arbor, MI, USA. [6]Munich Data Science Institute (MDSI), Technical University of Munich, Garching, Germany. [7]Department of Biochemistry, University of Oxford, Oxford, UK. [8]Department of Chemistry, University of Oxford, Oxford, UK. [9]These authors contributed equally: Nikita Levin, Cemil Can Saylan. ✉e-mail: Mathias.Wilhelm@tum.de; Shabaz.Mohammed@chem.ox.ac.uk

## Methods

### Chemicals, solvents, cell lines and proteases

Sodium dodecylsulfate (SDS), dimethylsulfoxide (DMSO), chloroacetamide (CAA), acetonitrile, ethanol, triethylammonium bicarbonate (TEAB), ethyl acetate, formic acid and urea were purchased from Sigma Aldrich. Tris(2-carboxyethyl)phosphine (TCEP), ultrapure water and Pierce BCA protein assay kit were obtained from Thermo Fisher Scientific. Carboxyl-coated magnetic beads were acquired from Cytiva. Proteases were purchased from: Wako (LysC), ImmunoPrecise Antibodies (LysN), Promega (trypsin and GluC) and Roche (chymotrypsin). Expi293F (human cell line based on HEK293) cells were purchased from Thermo Fisher. *Arabidopsis thaliana* Col-0 seedling material was kindly gifted by C. Griffiths from Rothamsted Research, UK. *Escherichia coli* (K12) samples were donated by W. Liu at The Rosalind Franklin Institute.

### Cell lysis

Expi293F or *E. coli* cell pellets were resuspended in 2% SDS, 50 mM TEAB (for proteolysis using trypsin, LysN and LysC) or 1% SDS, 50 mM TEAB (for proteolysis using chymotrypsin) or 6 M urea, 50 mM TEAB (for proteolysis using GluC) and sonicated using BioruptorPico (Diagenode) for 30 cycles (30 s on–off). Protein concentration was determined using BCA assay following the manufacturer's protocol. Proteins were then reduced and alkylated with 10 mM TCEP and 50 mM CAA for 30 min.

### Protein extraction

Ground *A. thaliana* seedlings were homogenized in 100 mM Tris buffer, pH 7.6, containing 4% SDS, 1x Protease inhibitor cocktail (Roche), 1x PhosStop (Roche), 10 mM TCEP, 50 mM CAA and 20 mg ml$^{-1}$ PVPP beads (Alfa Aesar) in an ice-cold ultrasonic water bath for 30 min. The homogenized extract was then incubated for 2 h in an orbital shaker (400 rpm) at 20 °C and clarified using two 10 min room-temperature centrifugation steps at 17,000 × g.

### Proteolysis

Proteins from Expi293F underwent proteolysis using either trypsin, chymotrypsin, LysC, LysN or GluC, whereas proteins from *A. thaliana* and *E. coli* were digested only with trypsin. In proteolysis using trypsin, LysC and LysN, proteins were digested using a modified single-pot solid-phase assisted sample preparation method (SP3) using carboxyl-coated magnetic beads at a 4:1 w/w bead : protein ratio[55,56]. Proteins were precipitated onto the beads using acetonitrile (80% final concentration) and incubated for 30 min with shaking. The beads were washed three times (four times for *A. thaliana* samples) with 80% ethanol, followed by three washes with 100% acetonitrile before incubation with digestion buffer (50 mM TEAB), containing a protease (trypsin, LysN or LysC) at a 1:25 enzyme : protein ratio for 4 h. Supernatant was collected, beads were washed with 2% DMSO solution, and the wash was combined with supernatant before acidification with formic acid to a 5% final concentration and centrifugation at 16,000 × g for 10 min. In proteolysis using chymotrypsin, protein samples were diluted twofold and digested with chymotrypsin at an enzyme : protein ratio of 1:50 for 4 h. Peptides were then acidified with formic acid and washed with saturated ethyl acetate five times[57]. In proteolysis using GluC, protein samples were diluted to adjust urea concentration to 0.8 M and digested with chymotrypsin at an enzyme : protein ratio of 1:35 for 4 h. As a final step in all protocols, peptides were desalted on Oasis HLB cartridges (Waters) and eluted with 50% acetonitrile in ultrapure water. The eluent containing peptides was dried using Genevac EZ-2.

### Ultra-high performance liquid chromatography fractionation[40]

Digests of Expi293F proteins were reconstituted in water containing 5% formic acid and 5% DMSO and processed using ultra-high performance liquid chromatography (UHPLC) fractionation in a Water Acquity Plus system. The peptides were separated using a Waters Acquity UPLC peptide CSH C18 (1 mm × 150 mm, 1.7 μm, 130 Å) column. The composition of solvent B (80% acetonitrile, 20% water) changed from 12% to 40% over 55 min and then from 40% to 50% over 5 min. Solvent A consisted of 10 mM TEAB (pH 8.0-8.2), 2% acetonitrile and 98% water. Each fraction was collected for 45 s (80 fractions in total). Fractions were concatenated as follows: fraction 1 was pooled with fractions 21, 41 and 61; other fractions were pooled in a similar fashion. Each pooled fraction contained approximately 6 μg of peptides in 200 μL solvents. Each pooled fraction was then split into 10 aliquots, each aliquot was further diluted by adding 80 μl 5% DMSO–5% formic acid to 20 μl peptide mixture.

### LC-MS data-dependent acquisition analyses

Liquid chromatography–tandem mass spectrometry (LC-MS/MS) data were acquired using an UltiMate 3000 nanoUHPLC system (Thermo Fisher Scientific) coupled either to an Orbitrap Exploris (Thermo Fisher Scientific) equipped with an Omnitrap (Fasmatech) for UVPD, EID, ECD and HCD analyses or to an Orbitrap Ascend Tribrid (Thermo Fisher Scientific) for ETciD analyses. The peptides were trapped on a C18 PepMap100 pre-column (300 μm (internal diameter, i.d.) × 5 mm, 100 Å, Thermo Fisher Scientific) using solvent A (0.1% formic acid in water), then separated on an in-house packed analytical column (50 μm (i.d.) × 50 cm in-house packed with ReproSil Gold 120 C18, 1.9 μm, Dr Maisch GmbH). The composition of solvent B (0.1% formic acid in acetonitrile) changed from 10% to 33% over 30 or 60 min for parameter optimization experiments and for HCD analyses, or from 8% to 28% over 120 min for UVPD, EID and ECD analyses. Full-scan MS1 spectra were acquired in the Orbitrap (scan range 400–1,300 *m/z*, resolution 60,000, automatic gain control (AGC) target 300%). In DDA of fractionated digests, the amount of material injected was 500 ng of each digest fraction except trypsin, for which 250 ng of each fraction were injected. For HCD analyses, the amount of material was further reduced by half for all proteases. The top 20 (40 in HCD) most abundant peptides were selected in each round of DDA for fragmentation. EID, UVPD and ECD were performed in the Omnitrap. Their products were mass-analyzed in the Orbitrap at 45,000 resolving power, the AGC was set to 200%, and the maximum injection time was set to 64 ms. ETciD was performed in the linear ion trap of the Orbitrap Ascend. Precursor ions underwent ETD for the following reaction times: 2+ and 3+ for 50 ms, 4+ for 25 ms, and 5+ to 7+ for 16 ms. Charge-reduced species were further activated using ion-trap CID at 35% of normalized collision energy. Products of ETciD were mass-analyzed in the Orbitrap at 7,500 resolving power, the AGC was set to 200%, and the maximum injection time was set to 64 ms. HCD was performed in the HCD cell of the Orbitrap Exploris. Products of HCD were mass-analyzed in the Orbitrap at 7,500 resolving power, the AGC was set to 40%, and the maximum injection time was set to 64 ms.

### LC-MS data-independent acquisition analyses

The DIA analyses were conducted in the same way as the DDA analyses, with the following adjustments: the composition of solvent B (0.1% formic acid in acetonitrile) changed from 8% to 20% over 240 min for UVPD, EID and ECD DIA, or from 8% to 28% over 120 min for HCD DIA at a flow rate of 100 nl min$^{-1}$. The amount of injected material was 400 ng unfractionated tryptic digest of Expi293F cells, 1200 ng tryptic digest of *A. thaliana* and 300 ng tryptic digest of *E. coli*. DIA scans were acquired using 4- or 8-*m/z* isolation windows in the 400–700 *m/z* range over 0–80 min, 500–700 *m/z* range over 80–160 min, and 600–900 *m/z* range over 160–240 min of a 240 min liquid chromatography gradient. Products of ECD, EID, HCD and UVPD were mass-analyzed in the Orbitrap at 60,000 resolving power, the AGC was set to 2,000%, and the maximum injection time was set to 50 ms.

## Database searches of data-dependent acquisition data

Spectra were searched using FragPipe (v22.0) MSFragger 4.1 (ref. 42) with standard 'close' search settings against Uniprot database (UPR_Homo sapiens_9606 accessed on 29 October 2024). The number of missed cleavages was set to four when searching GluC and chymotrypsin data. The types of fragments searched by MSFragger were set in accordance with fragmentation techniques used in the corresponding experiments. Ions $a + 1$, $c − 1$, $x + 1$ and $z + 1$ were manually defined for searches. MSBooster[29] and all quantification parameters were disabled in all searches including HCD data. Mass calibration and parameter optimizations were enabled. Data filtering was performed using Percolator (v3.6.5) and Philosopher (v5.1.1), followed by ProteinProphet. Results were reported using sequential FDR estimation, with a 1% FDR threshold at the protein level.

## Spectra annotation

Every raw file and its corresponding pepxml file generated by MSFragger were read and merged into a single dataframe, containing amino acid sequence, precursor charge, modifications (if any) and raw mass spectrum. For each PSM identified by MSFragger, its corresponding raw spectrum was annotated by matching with 10 ppm tolerance against a theoretical spectrum containing all ions with the following parameters: ion types $a$, $a + 1$, $b$, $c − 1$ $c$, $x$, $x + 1$, $y$, $z$, $z + 1$, fragment charges +1 to +3, and fragment lengths 1–29. See Supplementary Table 1 for ion type nomenclature. The annotation was carried out excluding all matches in a 2.4 Da window centered on the precursor ion's $m/z$. For additional quality control, we explicitly annotate the first four isotopic precursor peaks, for both the precursor charge and charge + 1, for the purpose of excluding them from a coincidental identification with another ion. Annotations for all PSMs are saved to one dataframe per fragmentation method, along with overall statistics enumerating the occurrences of each searched ion from our permutation.

## Creation of deep learning datasets

Datasets for the training of models were created as multiple parquet files from the annotated PSM datasets for all fragmentation methods. The PSM list was first cleaned of decoy sequences and sequences that contain no identified annotations. The output space–ion dictionary of the model was determined by including only annotated ions that occurred 100 or more times in the dataset. The PSMs were de-duplicated to unique sequence–charge–fragmentation method spectra, retaining the instance with the highest hyperscore from MSFragger. The resulting unique PSM dataset was then shuffled, split randomly into train–validation–test using a 0.8–0.1–0.1 split, and written into parquet files (see Figure SN1 and Table SN1 in Supplementary Notes for an evaluation of the appropriateness of this splitting strategy). For the purposes of multi-threading during dataloading, the training set was written to 8–12 separate files, all approximately having the same number of samples.

All dataset parquet files contain the original raw file name, scan number, modified peptide sequence, charge, fragmentation method, annotated ion strings and annotated ion intensities. The raw file names and scan numbers are sufficient to map any training–evaluation example back to its raw spectrum. Peptide sequence, charge and fragmentation method were the inputs to the model; the target outputs for training were constructed from the ion string and intensity. With a defined ion dictionary for the model, each ion string was mapped to its designated index, and its intensity was placed into a vector with the same length as the number of ions the model predicts (total of 815 fragment ions). For ions that were the same length or longer than the peptide, or for which the charge was greater than the precursor charge, a '−1' was assigned as the intensity and used to mask out these ions for training. All other unannotated ions were assigned an intensity of 0. All non-negative entries in the target spectrum vector were max-scaled by the most intense ion, such that the intensities fall between 0 and 1.

## Deep learning training and models

We used the original gated recurrent unit (GRU) cell recurrent neural network (RNN) of Prosit, but with specific architectural tweaks to accommodate the different metadata and the unordered structure of its output space. The metadata, that is, charge and fragmentation method, were both represented as one-hot vectors, concatenated and then linearly projected into 256 units; collision energy was not used as input for this model. Whereas the original Prosit model projected the metadata and sequence representation together multiplicatively, followed by replicating the combined representation into 174 identical vectors (Prosit's original output length), our sequence dimension retained the original maximum sequence length dimension (30 amino acids) and was multiplicatively combined with the 256-long metadata vector. This encoding was then passed to the RNN GRU cell bi-directional decoder layer. The final regressor then applied a linear transformation to our 815-ion output space, followed by a LeakyReLU activation function and mean pooling over the sequence dimension, yielding an 815-long vector of intensity predictions.

The model was trained using masked spectral distance. All indices of ions impossible to occur for a particular peptide were multiplied by 0 so that they do not contribute to the gradient during training. All models were trained for 30 epochs. The Adam optimizer was used with a learning rate of $1 \times 10^{-4}$, linearly warmed up for 20,000 steps at the beginning of training. Pytorch was used for implementation. The model is publicly available via the Koina prediction service under the title 'Prosit_2025_intensity_MultiFrag'.

## Rescoring with Oktoberfest

To assess rescoring under 1% FDR control at both the PSM and peptide levels, spectra were searched with varied parameters, using the same versions of FragPipe, MSFragger and the FASTA file as in the initial data analysis. The number of missed cleavages was maintained as previously established, with the following ion types specified according to the fragmentation method: $c$, $z$ for ECD, $a$, $b$, $c$, $x$, $y$, $z$ for EID, and $b$, $y$ for both HCD and UVPD. To ensure compatibility and reduce confounding factors in the comparison, we modified several parameters: mass calibration, parameter optimizations, $N$-terminal acetylation, MSBooster, ProteinProphet and all quantification settings were disabled, and the maximum peptide length was set to 30. The 'pin' files generated by FragPipe were concatenated and used for Percolator to obtain 1% FDR control at both the PSM and peptide levels. The results were subsequently used for downstream comparisons with the rescoring results.

Rescoring was conducted using Oktoberfest and Percolator (v3.6.5) on 100% FDR search results. No calibration for collision energy was applied, and indexed retention time (iRT) predictions were made using the 'Prosit_2019_irt' model. Intensity predictions were performed for the following ion fragmentations: $a + 1$, $b$, $c − 1$, $c$, $y$, $z$, $z + 1$ for ECD; $a$, $a + 1$, $b$, $c$, $y$, $x$, $x + 1$, $z$, $z + 1$ for EID; $a$, $b$, $y$ for HCD; and $a$, $a + 1$, $b$, $c$, $y$, $z$ for UVPD. Percolator was used to filter the data to 1% FDR at both the PSM and peptide levels.

## Database searches of data-independent acquisition data

Spectra were searched using FragPipe (v23.0) with MSFragger-DIA[58] (v4.1) with standard 'DIA_SpecLib_Quant' search settings against Uniprot databases ('UPR_Homo sapiens_9606' accessed on 22 August 2022, 'UPR_ArabidopsisThaliana_3702' accessed on 22 August 20222 and 'UPR_EscherichiaColi' accessed on 7 September 2022). For Expi293F (Proteome ID: UP000005640), FASTA contains one protein sequence per gene and contains a total of 20,702 protein entries (Swis-Prot 20,390 and TrEMBL 312 protein sequences). *Escherichia coli* (Proteome ID: UP000000625) FASTA contains a total of 4,489 protein entries (Swis-Prot 4,474 and TrEMBL 15 protein sequences), and *A. thaliana* (Proteome ID: UP000006548) FASTA contains a total of 39,443 protein entries (Swis-Prot 16,278 and TrEMBL 23,165 protein sequences). EasyPQP library generation and DIA-NN quantification were turned off.

Types of fragments searched by MSFragger were set in accordance with fragmentation techniques used in the corresponding experiments. Mass calibration and methionine oxidation variable modifications were turned on, and the maximum length of peptide was set to 30. The 'Prosit_2019_irt' and 'Prosit_2025_intensity_MultiFrag' models were used in MSBooster[29] via Koina[47] to predict peptide properties. Data filtering was performed using Percolator (v3.7.1) and Philosopher (v5.1.1), and results were reported at 1% PSM, 1% peptide and 1% protein levels.

## Reporting summary

Further information on research design is available in the Nature Portfolio Reporting Summary linked to this article.

## Data availability

Raw LC-MS data have been uploaded to the ProteomeXchange Consortium via the PRIDE partner repository with the dataset identifier PXD065289. Model, training, validation and test datasets can be found on Zenodo via the following link: https://doi.org/10.5281/zenodo.15755223 (ref. 59).

## Code availability

Source code and scripts are available on GitHub: Oktoberfest (https://github.com/wilhelm-lab/oktoberfest), Koina (https://github.com/wilhelm-lab/koina), MSBooster (https://github.com/Nesvilab/MSBooster) and the source code for training the Multifrag model (https://github.com/wilhelm-lab/Prosit_multifrag). Attribution for reused components and dependencies is provided in the repository.

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

## Acknowledgements

N.L, Y.D. and S.M. were supported by the EPSRC (V011359/1 (P)). J.S. and S.M. were supported by BBSRC responsive mode (BB/T016272/1). A.I.N. and K.L.Y. were supported in part by NIH grants R01-GM-094231 and U24-CA210967 (to A.I.N.) and a University of Michigan Rackham Predoctoral Fellowship (to K.L.Y.). J.L. and M.W. were, in part, supported by an ERC Starting Grant (Grant No. 101077037). C.C.S. was supported by the Elitenetzwerk Bayern (grant number F-6-M5613.6.K-NW-2021-411/1/1). The funders had no role in study design, data collection and analysis, decision to publish or preparation of the manuscript. The authors thank A. Makarov and K. Fort (both Thermo Fisher Scientific) and D. Papanastasiou, A. Smyrnakis and I. Orfanopoulos (all three at Fasmatech, Bruker Daltonics) for invaluable support. The authors are grateful to C. Griffiths from Rothamsted Research and W. Liu from The Rosalind Franklin Institute who kindly gifted *Arabidopsis thaliana* Col-0 seedling material and *Escherichia coli* samples, respectively.

## Author contributions

S.M. and M.W. conceived and supervised the project. N.L. and S.M. designed LC-MS experiments and wrote the original draft. N.L. optimized parameters of Omnitrap LC-MS, performed LC-MS experiments and initial data analysis and prepared the figures. C.C.S. performed comprehensive data analysis and data post-processing. J.L. wrote the code for and inspected the Prosit Multifrag model. Y.D. and J.S. prepared samples. K.L.Y. and A.I.N. implemented the Prosit Multifrag model into FragPipe MSBooster. S.M., M.W. and A.I.N. acquired funding. All authors revised the manuscript and approved the final version.

## Competing interests

A.I.N. is the Founder of Fragmatics and serves on the scientific advisory boards of Protai Bio, Infinitopes and Mobilion Systems. A.I.N. is also a paid consultant for Novartis. A.I.N. has a financial interest due to the licensing of MSFragger and IonQuant to commercial entities. M.W. is a founder and shareholder of MSAID GmbH with no operational role, and a member of the scientific advisory board of Momentum Biotechnologies. The other authors declare no competing interests.

## Additional information

**Extended data** are available for this paper at https://doi.org/10.1038/s41592-026-03042-9.

**Correspondence and requests for materials** should be addressed to Mathias Wilhelm or Shabaz Mohammed.

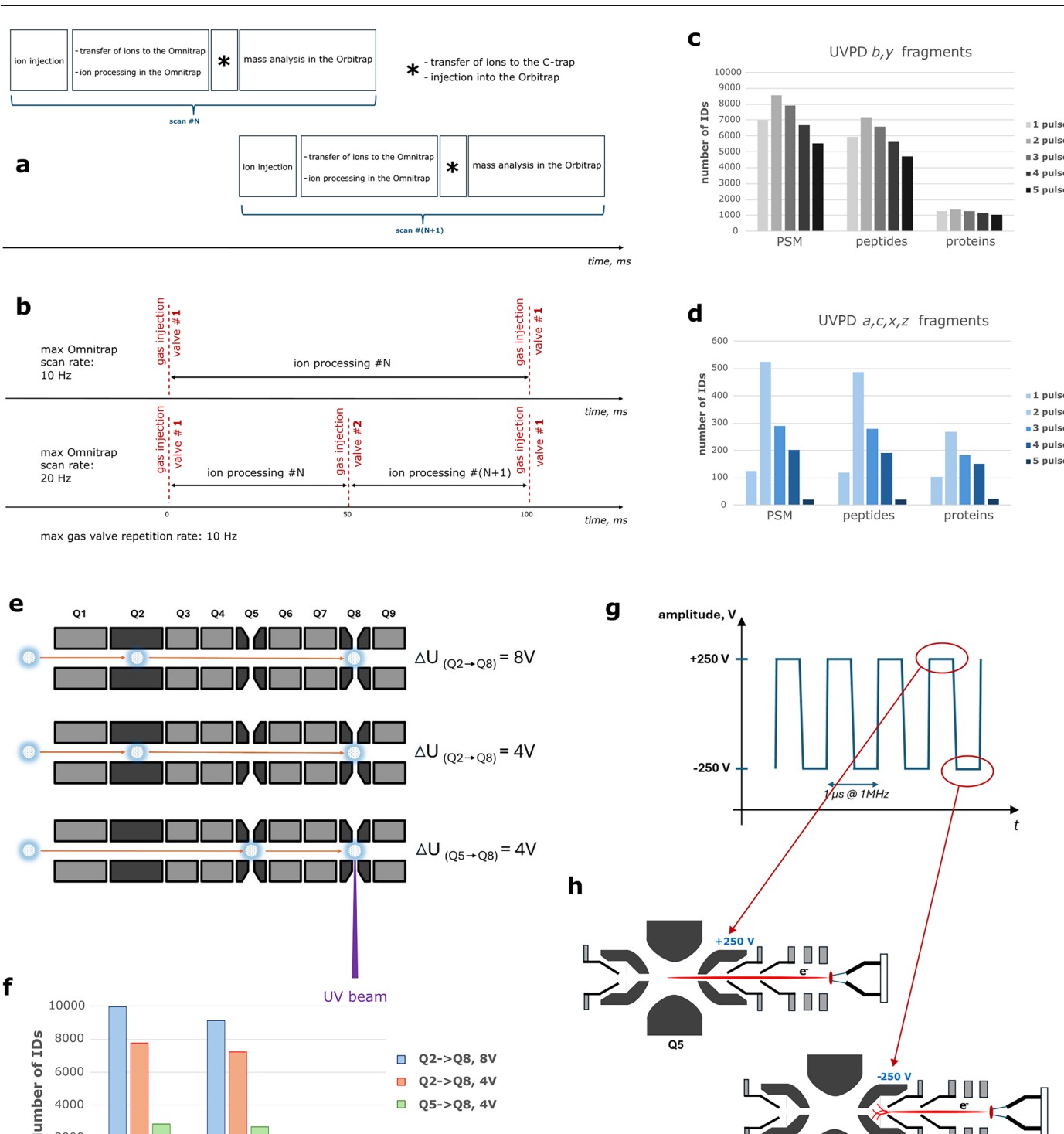

**Extended Data Fig. 1 | Ion processing workflow, gas pulsing, electron injection and UVPD parameters in the Omnitrap. a,** schematic workflow of ion processing in the Orbitrap-Omnitrap instrument; **b,** illustration of the minimum time distance between two injections of buffer gas into the Omnitrap using one valve (top) or two valves (bottom). The buffer gas is necessary for transfer, thermalization, and trapping of ions; **c-d,** Numbers of peptide–spectrum matches (PSMs), peptides and proteins identified in the UVPD LC-MS experiments using *b* and *y* fragments (**c**) or *a,c,x,z* fragments (**d**) for data analysis. 30 min LC gradients were used for the analysis of tryptic digest of human cells;

**e,** Schematic representations of different designs of ion transfer to Q8 segment of the Omnitrap. ΔU refers to the potential energy difference between two segments of the ion trap; **f,** numbers of PSMs, peptides and proteins identified in the LC-MS experiments using three different designs of ion transfer without UV laser triggering. *b* and *y* fragments were used for data analysis; **g,** time scan of the electric potential at the Q5 electrode, where the source of electrons is installed; **h,** schematic of the injection of electrons into the Q5 segment of the Omnitrap during positive (top) and negative (bottom) RF phases.

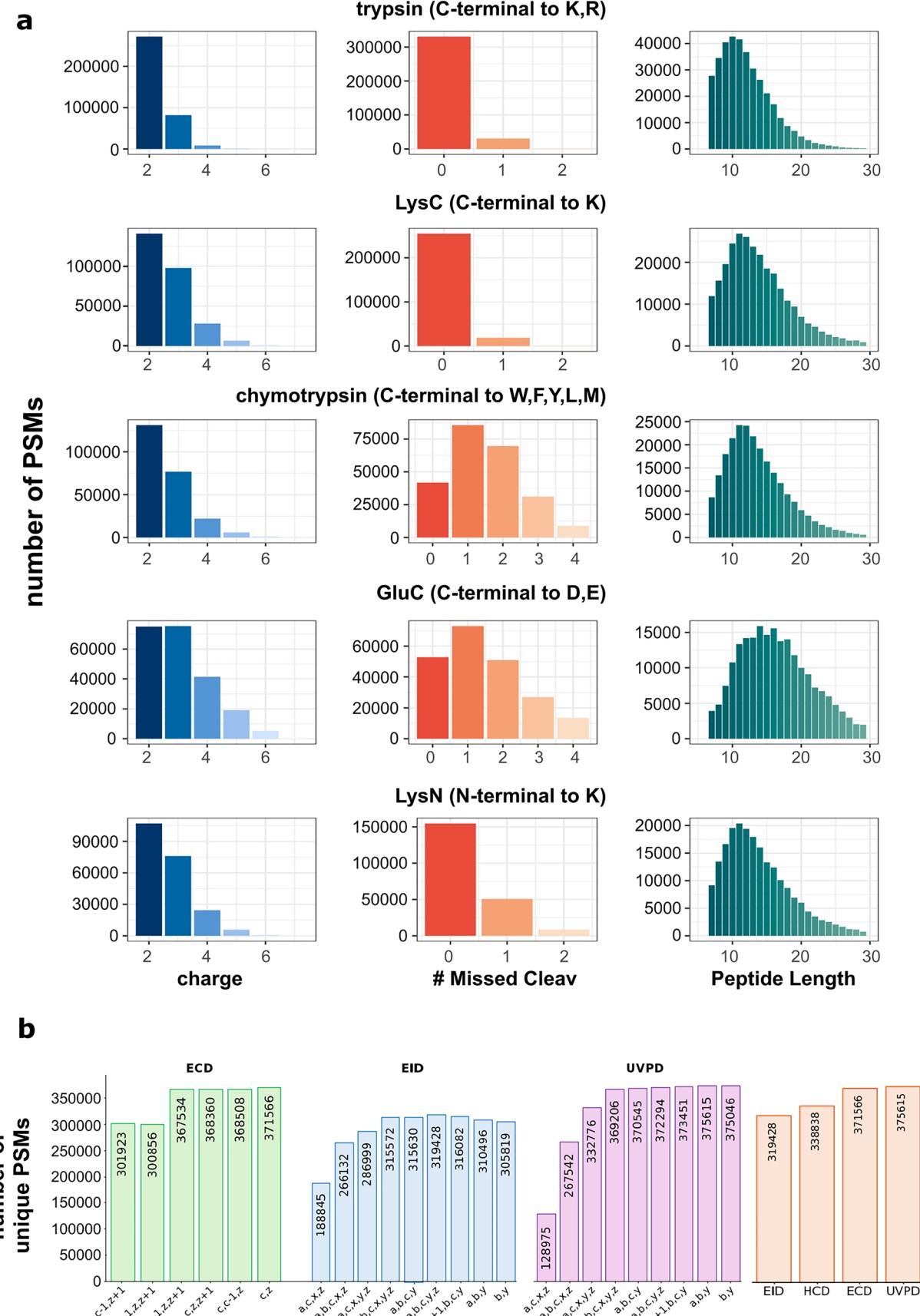

**Extended Data Fig. 2 | Enzyme-specific physicochemical properties and unique PSMs in ECD, EID and UVPD data. a**, Per-enzyme distributions of charge states (left), missed cleavages (middle) and peptide lengths (right) for all PSMs identified in 2D-LCMS-HCD experiments; **b**, total numbers of unique peptide–spectrum matches (unique combination of amino acid sequence, charge, and modification) identified using different combinations of fragment types in ECD, EID, and UVPD experiments; **c**, highest number of PSMs from **b** together with the HCD data.

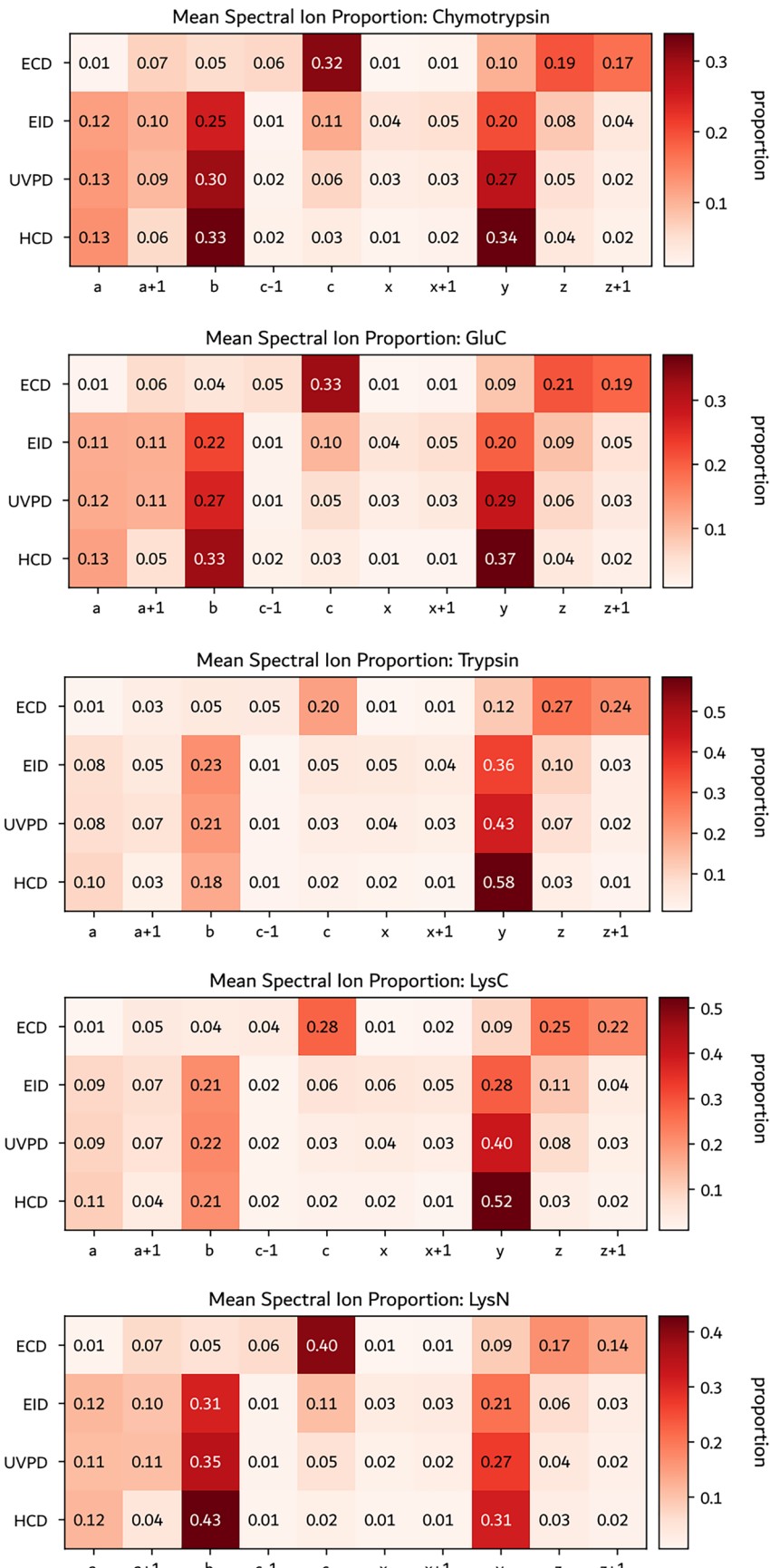

**Extended Data Fig. 3 | Averaged frequencies of ions in ECD, EID and UVPD data per enzyme.** Heatmap of average proportions of fragment ion peaks of different types among all annotated peaks in ECD, EID, HCD, and UVPD spectra plotted per enzyme, not reflecting relative ion intensities. Annotation was performed for 10 ion types: $a, a+1, b, c-1, c, x, x+1, y, z, z+1$.

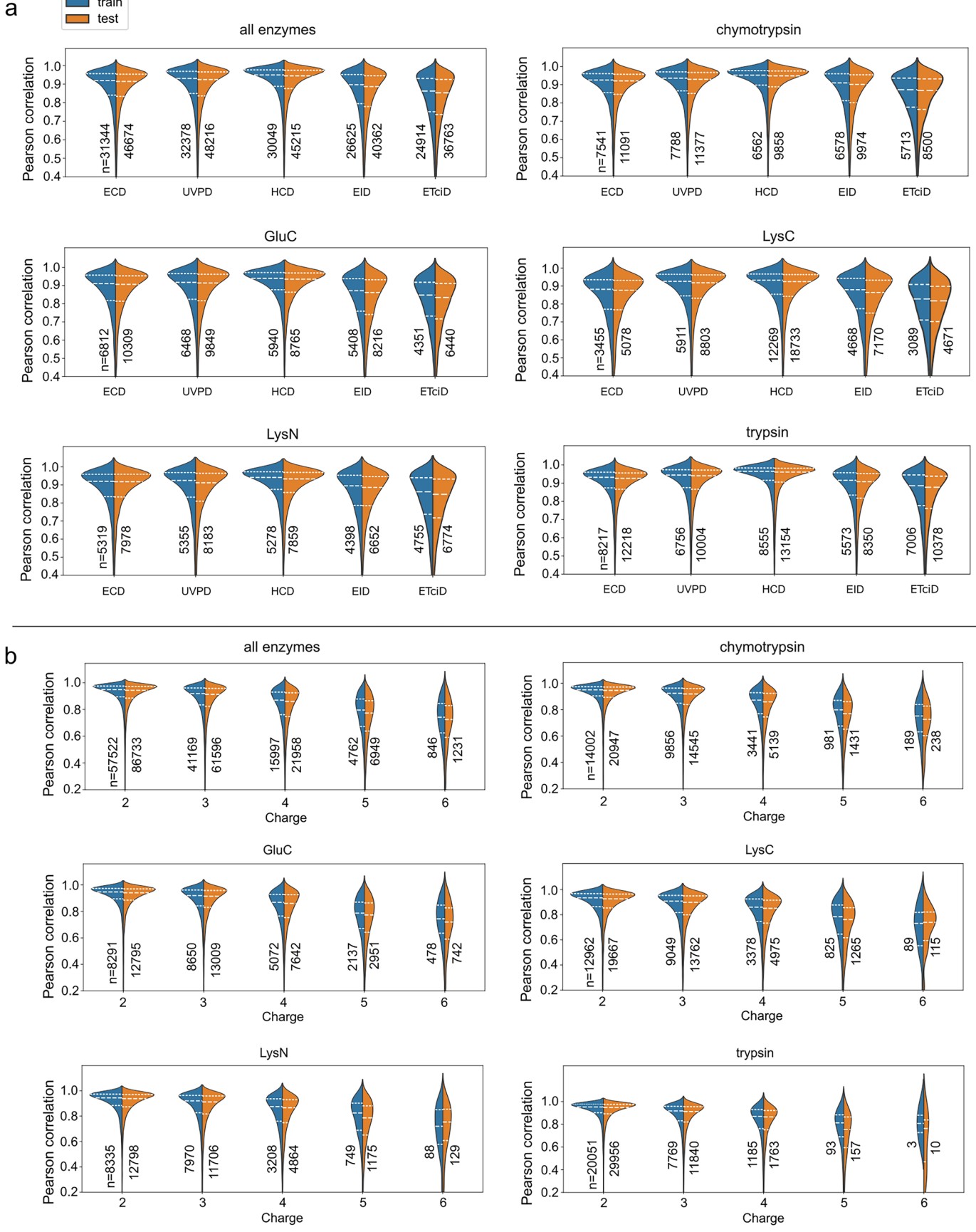

**Extended Data Fig. 4 | Pearson correlations per enzyme broken down per fragmentation technique and per charge state.** Distributions of Pearson correlation between experimental and predicted fragmentation spectra plotted for train and test datasets per enzyme per fragmentation method (**a**) or per enzyme per charge state (**b**). Horizontal dashed lines correspond to 25, 50, and 75% percentiles. Numbers (n) indicate sample sizes. Distributions protruding beyond 1.0 are plotting artefacts.

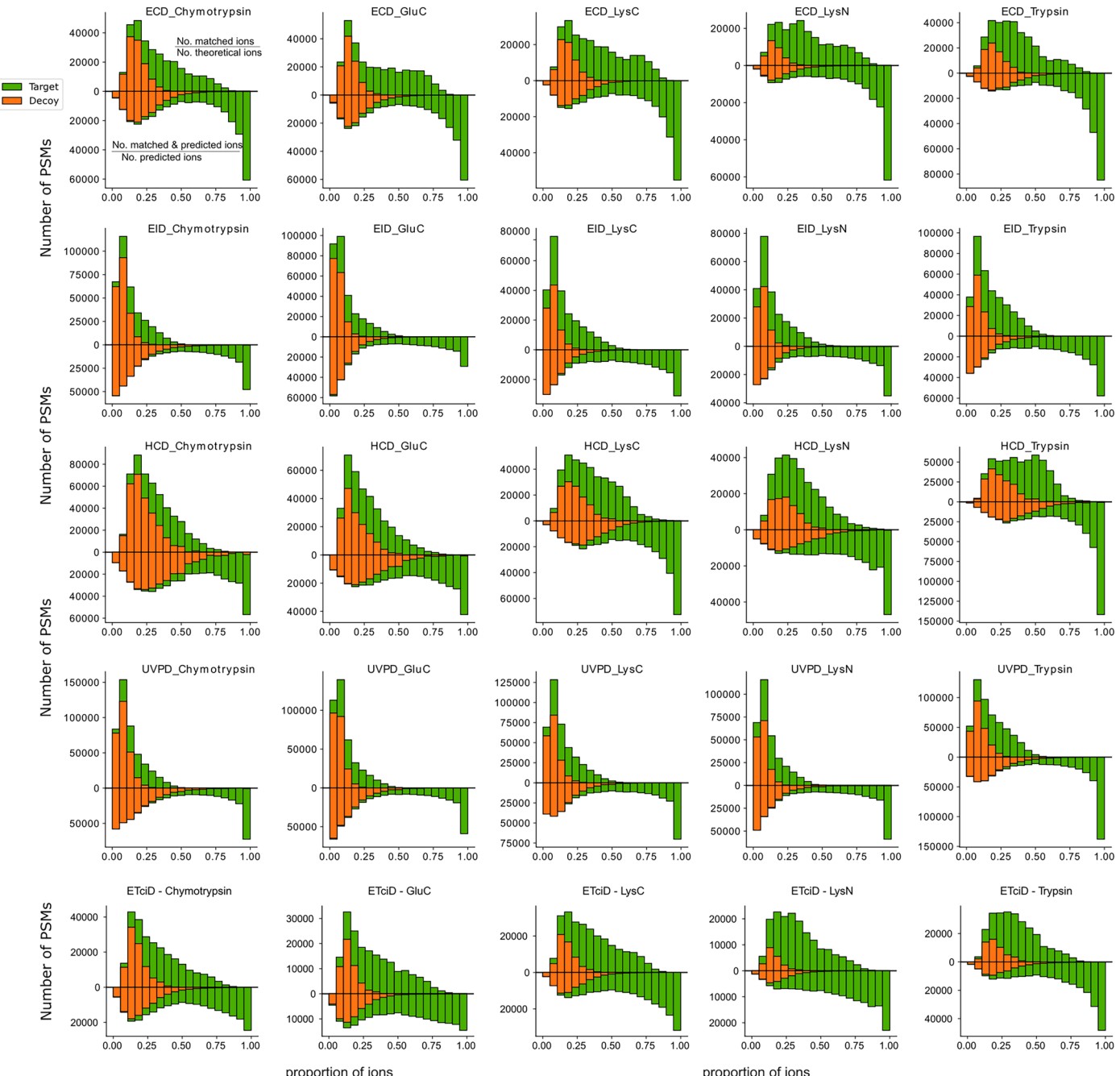

**Extended Data Fig. 5 | Predictions of fragment's presence or absence in a spectrum help separate target PSMs from decoy.** Histograms of the ratios of experimentally observed ions to all theoretically possible fragments (upper distributions); histograms of the ratios of experimentally observed and predicted ions to all predicted ions (bottom distributions). Target and decoy populations are shown in green and orange, respectively.

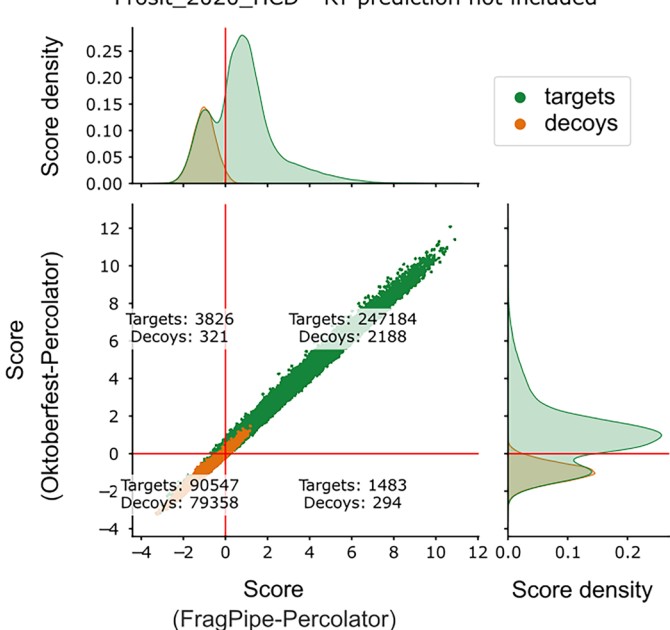

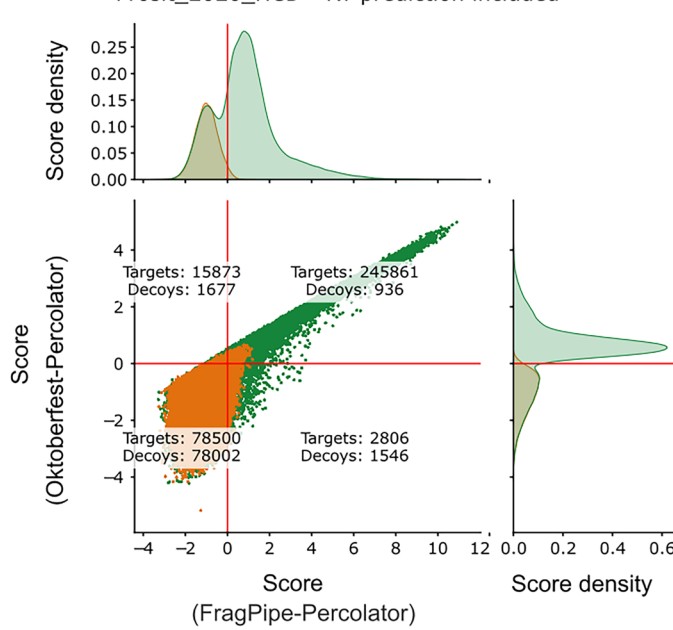

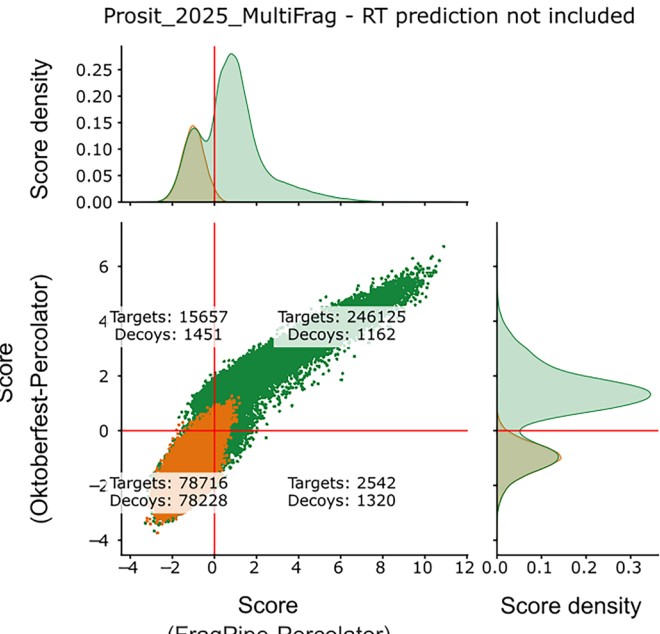

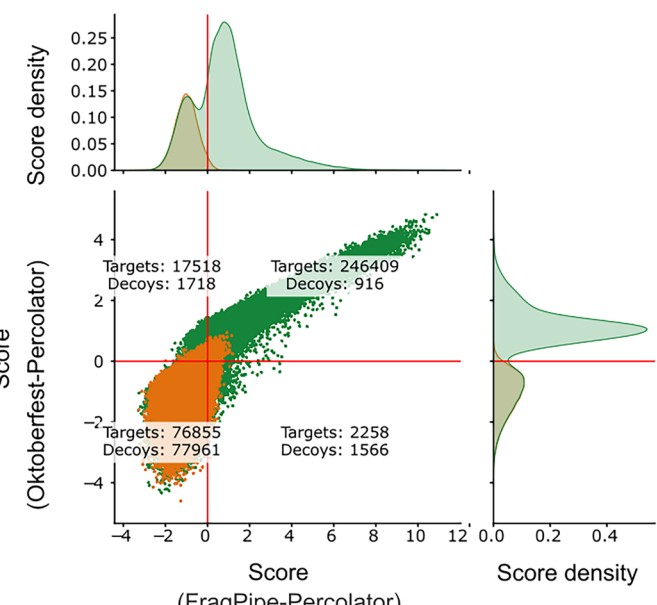

**Extended Data Fig. 6 | Effect of predictions of retention time and intensities of fragments.** Correlation of Percolator scores for all target (green) and decoy (orange) PSMs in trypsin ECD data acquired from rescoring the MSFragger (top) and Oktoberfest (right) sets of scores plotted per enzyme. The red lines indicate the 1% PSM-level FDR cut-offs in MSFragger and Oktoberfest score distributions.

The following models were used for rescoring: **top-left**, Prosit_2020_HCD (Ref. 27) without predictions of retention times (RT); **top-right**, Prosit_2020_HCD including predictions of retention times; **bottom-left**, Prosit_2025_MultiFrag without predictions of retention times; **bottom-right**, Prosit_2025_MultiFrag including predictions of retention times.

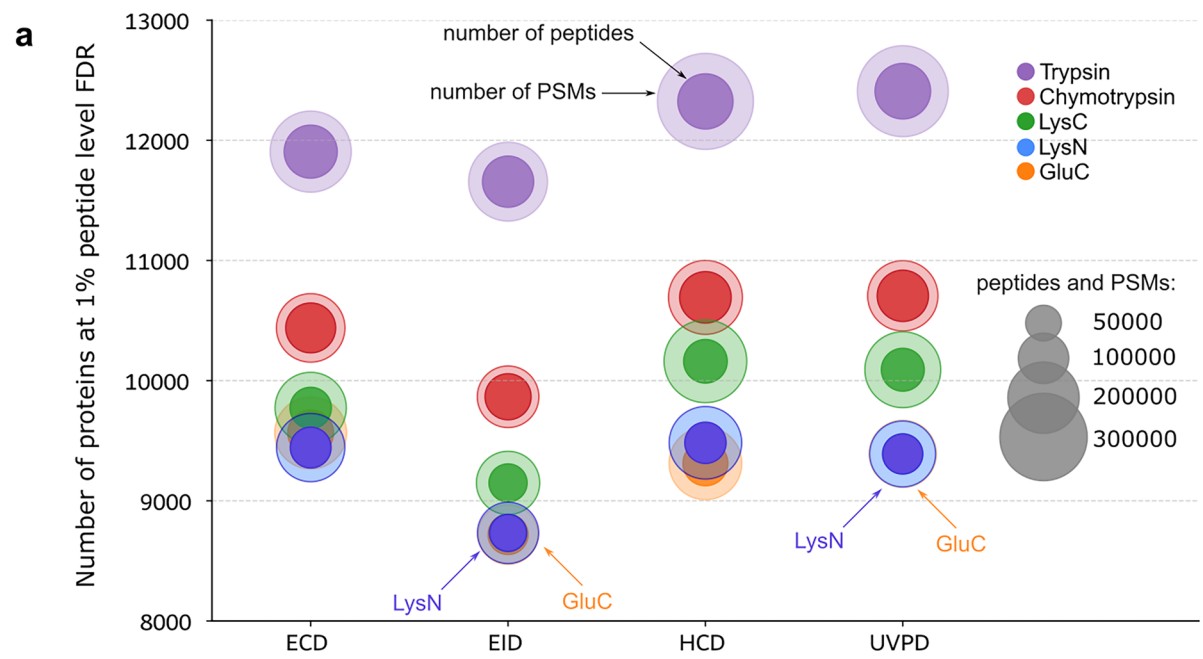

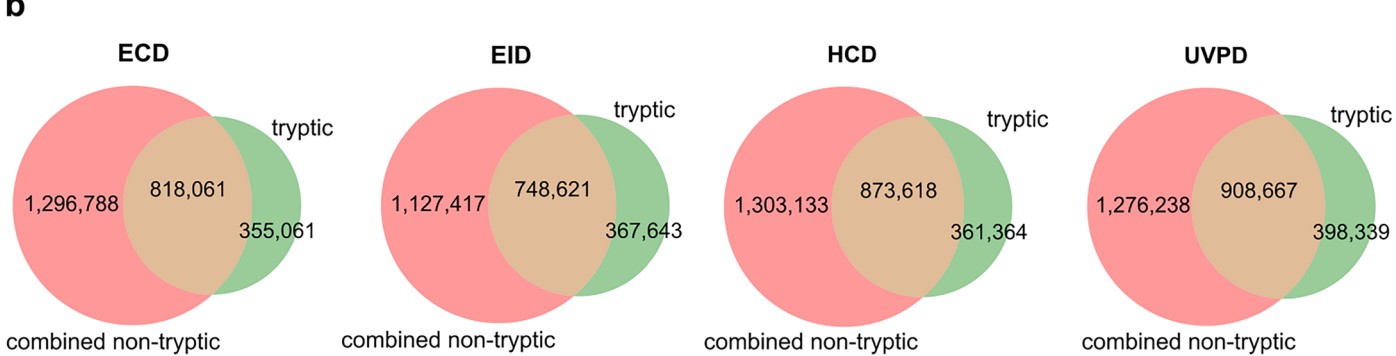

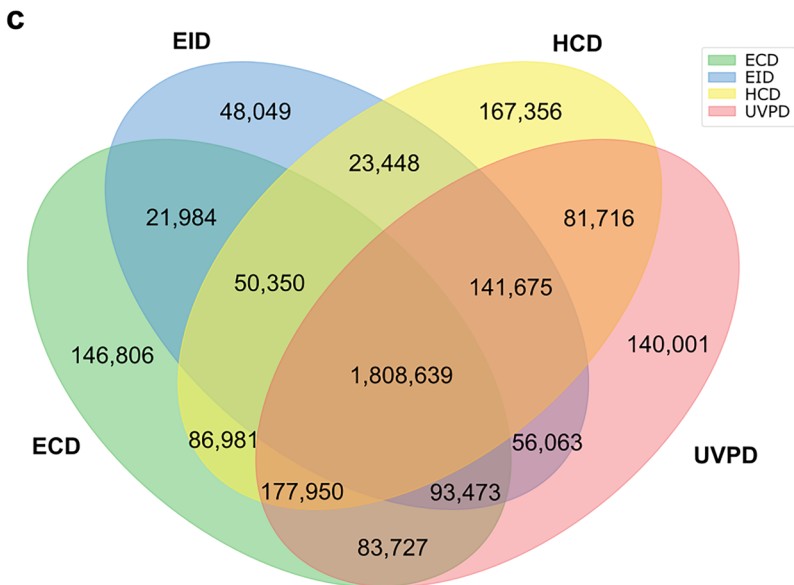

**Extended Data Fig. 7 | Relative performance of enzymes and fragmentation techniques in DDA data. a**, Numbers of PSMs (outer circles), peptides (inner circles) and proteins (y-axis) acquired using different fragmentation techniques and enzymes. **b**, Venn diagrams of total numbers of unique amino acids observed in tryptic datasets (green) and all non-tryptic datasets combined (red) using different fragmentation techniques. **c**, Venn diagram of unique amino acids observed using different fragmentation techniques across all enzymatic datasets.

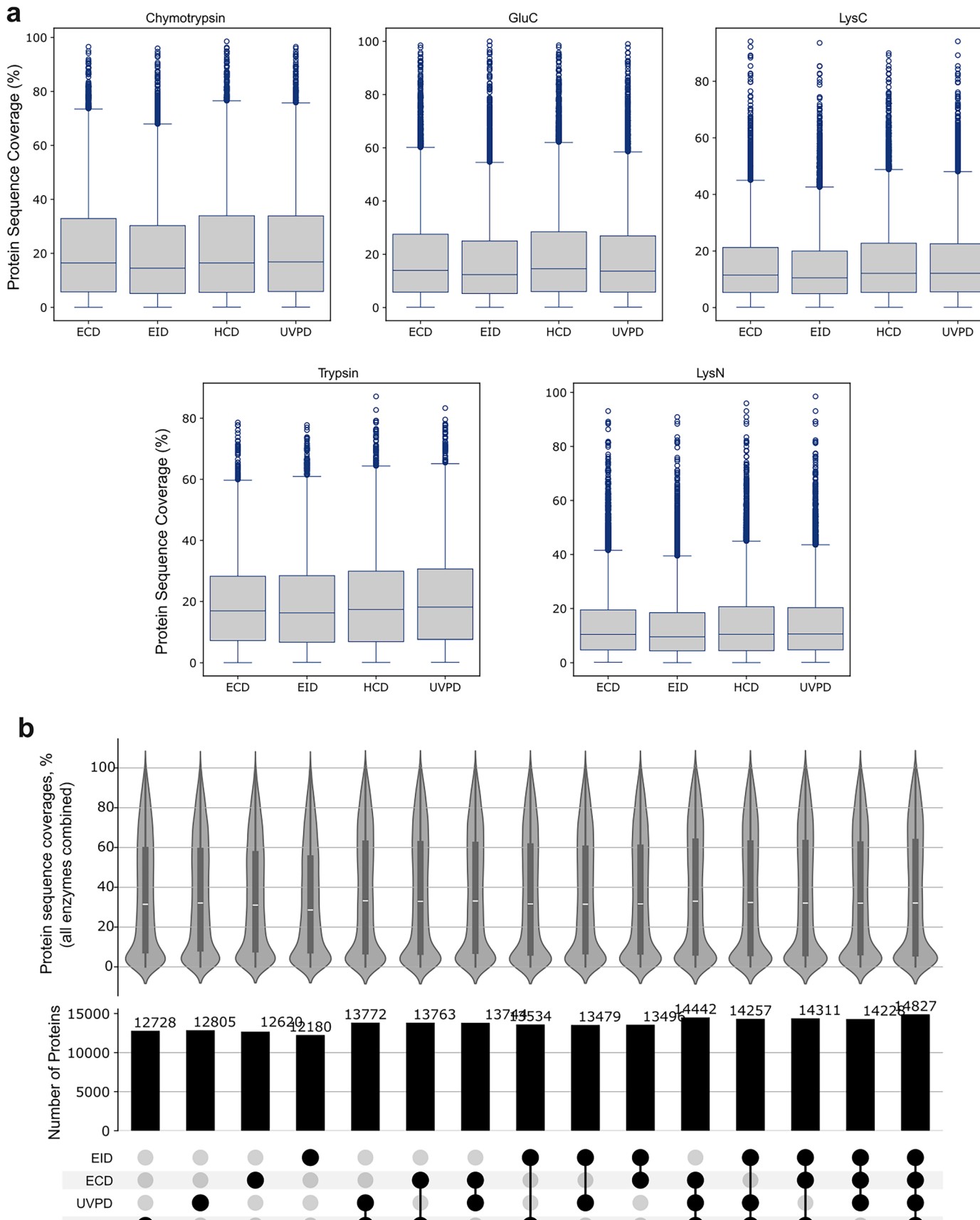

**Extended Data Fig. 8 | Proteome coverages in DDA data. a**, Box plots showing the distributions of protein sequence coverages of all proteins identified in different enzyme datasets using different fragmentation techniques. **b**, Upset plots showing the distributions of protein sequence coverages of all proteins in all enzymes combined in various combinations of fragmentation methods.

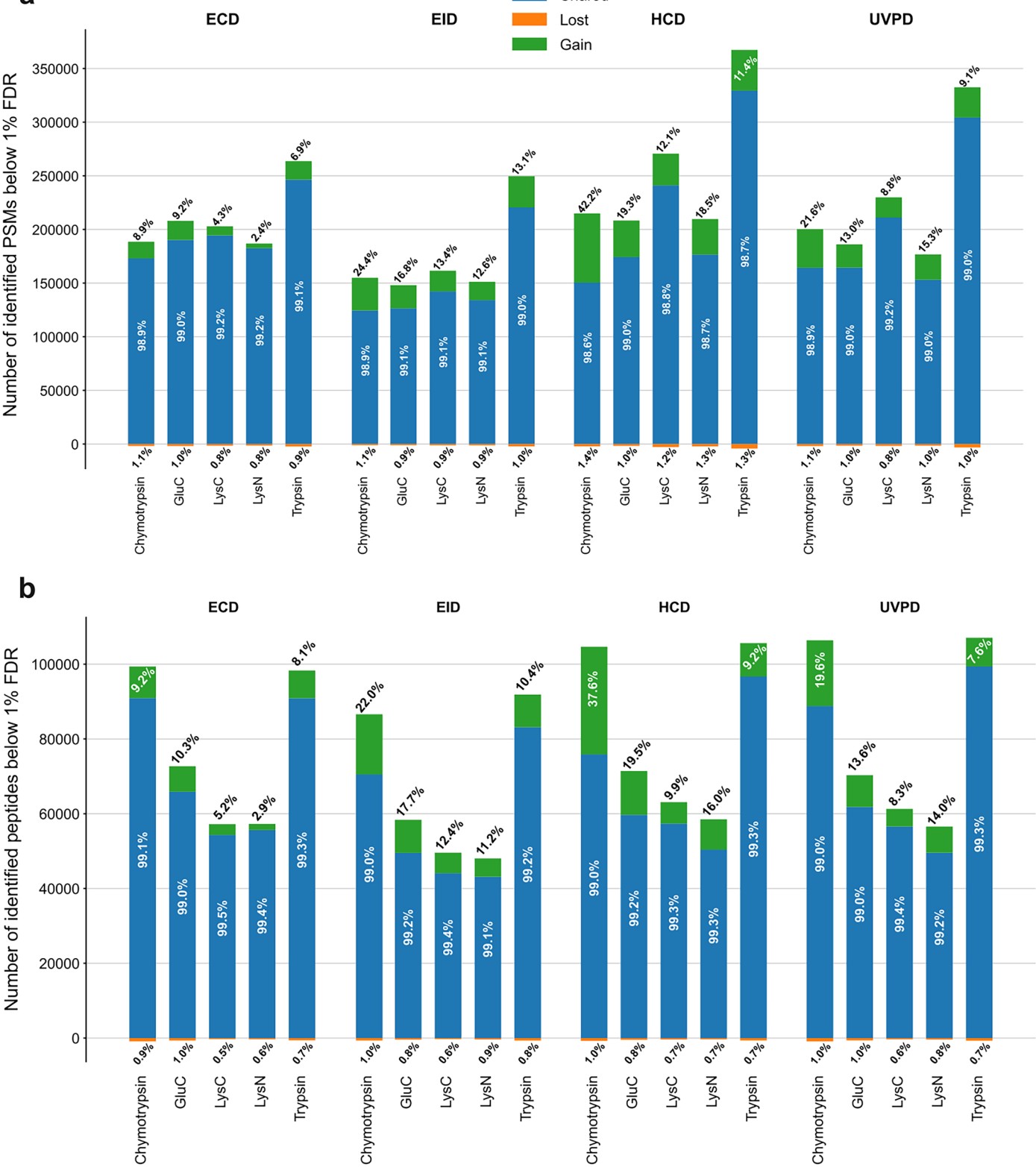

**Extended Data Fig. 9 | Gain/share/loss at the PSM and peptide levels (MSFragger with and without MSBooster).** Numbers of shared (blue), gained (green), and lost (orange) PSMs (**a**) and peptides (**b**) identified at 1% FDR using MSFragger with MSBooster equipped with the fragmentation prediction model compared to the original MSFragger searches (without MSBooster) for each fragmentation technique per enzyme.

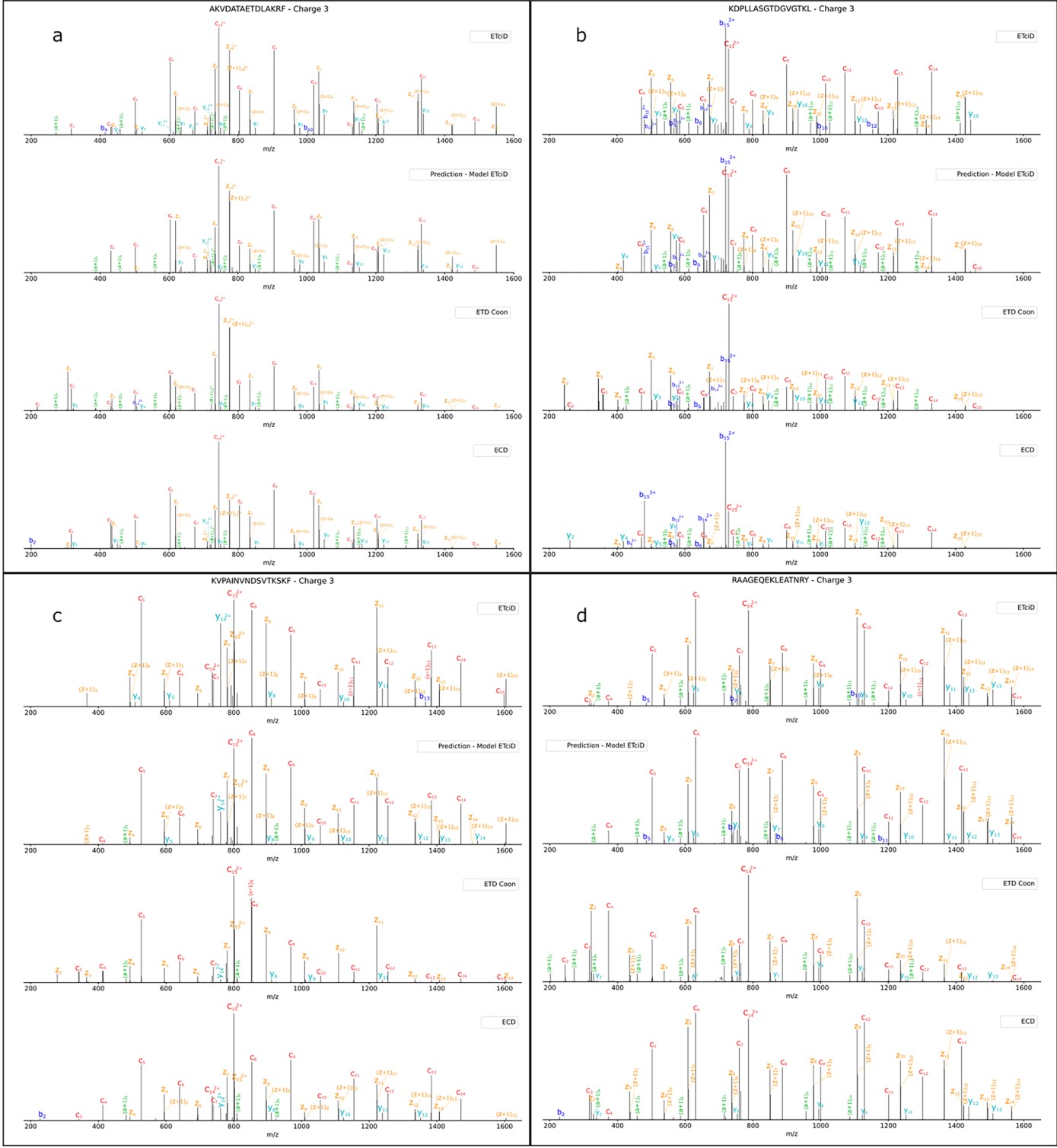

**Extended Data Fig. 10 | Comparison of ETciD, ECD and ETD spectra of four triply charged peptides from different datasets.** Comparison among spectra of triply charged AKVDATAETDLAKRF (**a**), KDPLLASGTDGVGTKL (**b**), KVPAINVNDSVTKSKF (**c**) and RAAGEQEKLEATNRY (**d**) peptide. Each panel contains ETciD spectrum acquired on Ascend Tribrid (this work), ETciD spectrum modeled by Prosit_Multifrag model, ETD spectrum acquired on Ascend Tribrid (Sinitcyn et al., Ref. 39), ECD spectrum acquired on Exploris-Omnitrap (this work). PCC number is the Pearson correlation coefficient between the spectrum and the ETciD prediction.

| | |
|---|---|

# Reporting Summary

## Statistics

For all statistical analyses, confirm that the following items are present in the figure legend, table legend, main text, or Methods section.

| n/a | Confirmed | |
|---|---|---|
| ☐ | ☒ | The exact sample size (*n*) for each experimental group/condition, given as a discrete number and unit of measurement |
| ☐ | ☒ | A statement on whether measurements were taken from distinct samples or whether the same sample was measured repeatedly |
| ☒ | ☐ | The statistical test(s) used AND whether they are one- or two-sided<br>*Only common tests should be described solely by name; describe more complex techniques in the Methods section.* |
| ☒ | ☐ | A description of all covariates tested |
| ☒ | ☐ | A description of any assumptions or corrections, such as tests of normality and adjustment for multiple comparisons |
| ☒ | ☐ | A full description of the statistical parameters including central tendency (e.g. means) or other basic estimates (e.g. regression coefficient) AND variation (e.g. standard deviation) or associated estimates of uncertainty (e.g. confidence intervals) |
| ☒ | ☐ | For null hypothesis testing, the test statistic (e.g. *F*, *t*, *r*) with confidence intervals, effect sizes, degrees of freedom and *P* value noted<br>*Give P values as exact values whenever suitable.* |
| ☒ | ☐ | For Bayesian analysis, information on the choice of priors and Markov chain Monte Carlo settings |
| ☒ | ☐ | For hierarchical and complex designs, identification of the appropriate level for tests and full reporting of outcomes |
| ☐ | ☒ | Estimates of effect sizes (e.g. Cohen's *d*, Pearson's *r*), indicating how they were calculated |

*Our web collection on statistics for biologists contains articles on many of the points above.*

## Software and code

Policy information about availability of computer code

| Data collection | LCMS spectra were collected on a custom Orbitrap Exploris-Omnitrap instrument and on an Orbitrap Ascend Tribrid instrument. After acquisition, data were processed in MSFragger (v4.1) within FragPipe (v22.0 or v23.0). |
|---|---|
| Data analysis | The "MultiFrag" model presented in the article was coded in Python using Pytorch v2.6.0 library. The training code for this model is available on Github (https://github.com/wilhelm-lab/Prosit_multifrag) and as a Koina instance at https://koina.proteomicsdb.org/. The rescoring of the initial MSFragger searches was performed in Oktoberfest (https://github.com/wilhelm-lab/oktoberfest) and MSBooster (v1.3.10) via FragPipe (v23.0). FDR estimations were done using Percolator (v3.6.5 and v3.7.1) and Philosopher (v5.1.1). Downstream data processing was done using Python 3.10. Python 3.10, R (4.2.2), Inkscape (v1.4) were used to prepare figures. |

For manuscripts utilizing custom algorithms or software that are central to the research but not yet described in published literature, software must be made available to editors and reviewers. We strongly encourage code deposition in a community repository (e.g. GitHub). See the Nature Portfolio guidelines for submitting code & software for further information.

## Data

Policy information about availability of data

All manuscripts must include a data availability statement. This statement should provide the following information, where applicable:
- Accession codes, unique identifiers, or web links for publicly available datasets
- A description of any restrictions on data availability
- For clinical datasets or third party data, please ensure that the statement adheres to our policy

> The LCMS proteomics data, database MSFragger searches and results of data rescoring were deposited to the ProteomeXchange Consortium via the PRIDE partner repository with the dataset identifier PXD065289. Model weights, training, test, and holdout data sets were deposited to Zenodo (doi:10.5281/zenodo.15755223 and https://zenodo.org/records/15064340)

## Research involving human participants, their data, or biological material

Policy information about studies with human participants or human data. See also policy information about sex, gender (identity/presentation), and sexual orientation and race, ethnicity and racism.

| | |
|---|---|
| Reporting on sex and gender | n/a |
| Reporting on race, ethnicity, or other socially relevant groupings | n/a |
| Population characteristics | n/a |
| Recruitment | n/a |
| Ethics oversight | n/a |

Note that full information on the approval of the study protocol must also be provided in the manuscript.

# Field-specific reporting

Please select the one below that is the best fit for your research. If you are not sure, read the appropriate sections before making your selection.

☒ Life sciences    ☐ Behavioural & social sciences    ☐ Ecological, evolutionary & environmental sciences

For a reference copy of the document with all sections, see nature.com/documents/nr-reporting-summary-flat.pdf

# Life sciences study design

All studies must disclose on these points even when the disclosure is negative.

| | |
|---|---|
| Sample size | n/a |
| Data exclusions | n/a |
| Replication | n/a |
| Randomization | n/a |
| Blinding | n/a |

# Reporting for specific materials, systems and methods

We require information from authors about some types of materials, experimental systems and methods used in many studies. Here, indicate whether each material, system or method listed is relevant to your study. If you are not sure if a list item applies to your research, read the appropriate section before selecting a response.

## Materials & experimental systems

| n/a | Involved in the study |
|-----|----------------------|
| ☒ | ☐ Antibodies |
| ☐ | ☒ Eukaryotic cell lines |
| ☒ | ☐ Palaeontology and archaeology |
| ☒ | ☐ Animals and other organisms |
| ☒ | ☐ Clinical data |
| ☒ | ☐ Dual use research of concern |
| ☐ | ☒ Plants |

## Methods

| n/a | Involved in the study |
|-----|----------------------|
| ☒ | ☐ ChIP-seq |
| ☒ | ☐ Flow cytometry |
| ☒ | ☐ MRI-based neuroimaging |

# Eukaryotic cell lines

Policy information about cell lines and Sex and Gender in Research

| | |
|---|---|
| Cell line source(s) | Expi293F Cells were purchased from Thermo Fisher Scientific. Expi293F human cells are derived from the 293 cell line and are a core component of the Expi293 Expression System. |
| Authentication | n/a; no further authentication was performed in-house. |
| Mycoplasma contamination | The cell line was specified negative for mycoplasma by the vendor in a mycoplasma qPCR assay |
| Commonly misidentified lines (See ICLAC register) | No commonly misidentified cell lines were used in this study. |

# Dual use research of concern

Policy information about dual use research of concern

## Hazards

Could the accidental, deliberate or reckless misuse of agents or technologies generated in the work, or the application of information presented in the manuscript, pose a threat to:

| No | Yes | |
|----|-----|---|
| ☒ | ☐ | Public health |
| ☒ | ☐ | National security |
| ☒ | ☐ | Crops and/or livestock |
| ☒ | ☐ | Ecosystems |
| ☒ | ☐ | Any other significant area |

## Experiments of concern

Does the work involve any of these experiments of concern:

| No | Yes | |
|----|-----|---|
| ☒ | ☐ | Demonstrate how to render a vaccine ineffective |
| ☒ | ☐ | Confer resistance to therapeutically useful antibiotics or antiviral agents |
| ☒ | ☐ | Enhance the virulence of a pathogen or render a nonpathogen virulent |
| ☒ | ☐ | Increase transmissibility of a pathogen |
| ☒ | ☐ | Alter the host range of a pathogen |
| ☒ | ☐ | Enable evasion of diagnostic/detection modalities |
| ☒ | ☐ | Enable the weaponization of a biological agent or toxin |
| ☒ | ☐ | Any other potentially harmful combination of experiments and agents |

## Plants

| | |
|---|---|
| Seed stocks | Arabidopsis thaliana Col-0 seedling material was gifted by Cara Griffiths  from Rothamsted Research |
| Novel plant genotypes | n/a |
| Authentication | not performed |

