## [Peer Review File · Nature Methods]

Integrating Alternative Fragmentation Techniques into Standard LC-MS Workflows Using a Single Deep Learning Model Enhances Proteome Coverage

Corresponding Author: Professor Shabaz Mohammed

Version 0:

Decision Letter:

7th Aug 2025

Dear Shabaz,

Your Article, "Integrating Alternative Fragmentation Techniques into Standard LC-MS Workflows Using a Single Deep Learning Model Enhances Proteome Coverage", has now been seen by 3 reviewers. As you will see from their comments below, although the reviewers find your work of considerable potential interest, they have raised a number of concerns. We continue to be interested in potentially publishing your manuscript in Nature Methods, but would like to evaluate your response to these concerns before we reach a final decision on publication.

We therefore invite you to revise your manuscript to address all the technical concerns that were raised by the reviewers.

*Editor's note: We apologize that due to a mix-up, the file that was made available to the reviewers was the originally submitted file that did not include information about data/code. Of course, I will also let the reviewers know of this when we send the revised paper back for review.

I am happy to chat with you, if you please, regarding the point reviewer #1 raised about splitting the paper in two. While we don't think this is necessary, and that further analysis and reorganization of the paper might be sufficient to focus the paper on the most relevant developments, I am happy to hear your thoughts on this.

- * include a point-by-point response to the reviewers and to any editorial suggestions
- * please underline/highlight any additions to the text or areas with other significant changes to facilitate review of the revised manuscript
- * address the points listed described below to conform to our open science requirements
- * ensure it complies with our general format requirements as set out in our guide to authors at www.nature.com/naturemethods
- * resubmit all the necessary files by using the link below to access your home page

Link Redacted

Note: This URL links to your confidential account and associated information about manuscripts you may have submitted, or that you are reviewing for us. If you wish to forward this email to co-authors, please delete this link.

We hope to receive your revised paper within 8 weeks. If you are substantially delayed, please let us know. In this event, we will still be happy to reconsider your paper at a later date so long as nothing similar has been accepted for publication at Nature Methods or published elsewhere.

OPEN SCIENCE REQUIREMENTS

REPORTING SUMMARY

When revising your manuscript, please update your reporting summary.

For any revision that includes light microscopy data, we ask our authors to please include a completed light microscopy reporting table [https://www.nature.com/documents/Light_microscopy_reporting_table.xlsx] to ensure the methods are described thoroughly. The table will be available to reviewers and ultimately published should the manuscript be accepted at the journal.

IMAGE INTEGRITY

EXTENDED DATA FIGURES

DATA AVAILABILITY

All novel DNA and RNA sequencing data, protein sequences, genetic polymorphisms, linked genotype and phenotype data, gene expression data, macromolecular structures, and proteomics data must be deposited in a publicly accessible database, and accession codes and associated hyperlinks must be provided in the "Data Availability" section.

For papers containing bioimaging data, we strongly recommend depositing the data to Bioimage Archive (<https://www.ebi.ac.uk/bioimage-archive/>). Associated accession codes and hyperlinks should be provided in the "Data Availability" section.

CODE AVAILABILITY

Please include a "Code Availability" subsection in the Online Methods which details how your custom code is made available. Only in rare cases (where code is not central to the main conclusions of the paper) is the statement "available upon request" allowed (and reasons should be specified).

MATERIALS AVAILABILITY

SUPPLEMENTARY PROTOCOL

To help facilitate reproducibility and uptake of your method, we ask you to prepare a step-by-step Supplementary Protocol for the method described in this paper. We [encourage authors to share their step-by-step experimental protocols](https://www.nature.com/nature-research/editorial-policies/reporting-standards#protocols) on a protocol sharing platform of their choice and report the protocol DOI in the reference list. Nature Portfolio's protocols.io is a free-to-use and open resource for protocols; protocols deposited onto protocols.io are citable and can be linked from the published article. More details can be found at [protocols.io](https://www.protocols.io/help/publish-articles).

ORCID

Please do not hesitate to contact me if you have any questions or would like to discuss these revisions further. We look forward to seeing your revised manuscript and thank you for the opportunity to consider your work.

Best regards,
Arunima

Arunima Singh, Ph.D.
Senior Editor

Reviewers' Comments:

Reviewer #1 (Remarks to the Author):

A dataset was generated using the Omnitrap platform interfaced to an Orbitrap and its diverse ion activation methods. To enhance the study's scope and relevance, a variety of specific proteases were employed to digest proteins. Although chymotrypsin is often regarded as a non-specific protease, Promega has demonstrated—through recent marketing literature—that a highly purified version exhibits substantial specificity. What makes the use of these “alternative” proteases particularly interesting is their relative neglect within the proteomics field. This is largely due to the fact that they do not yield the clean fragmentation patterns typically produced by peptides digested with trypsin. Nevertheless, their strategic utility is undeniable. For example, Josh Coon's lab published a study in *Nature Biotechnology* showing that the use of multiple proteases enabled high sequence coverage across various cell lines. Even so, the effort plateaued at approximately 80% sequence coverage—highlighting that even combinatorial use of proteases has limitations to achieve 100% sequence coverage. In my view, the field's reluctance to adopt these proteases stems from the increased complexity they introduce in database searching. Therefore, a detailed investigation of the fragmentation patterns generated by these proteases under various ion activation conditions would be a valuable contribution to the field. Such a study could illuminate how different fragmentation methods preferentially target specific amino acid binds for fragmentation, especially in the absence of strongly basic residues within a peptide sequence. This could form the basis of a compelling manuscript, provided sufficient detail is included to illustrate how fragmentation characteristics correlate with peptide composition. A separate paper focused on improved methods for identifying peptides generated by these proteases—along with a thorough description of the computational approach—would also be of substantial value to the community. Combined, these two interesting topics are diluted.

The study is well done, and the paper is clear and well written. They report a 10% improvement in peptide identifications which is not much of an advance—it may be better characterized as experimental variation. A more insightful metric might be the gain in sequence coverage, as demonstrated by the Coon study by combining the results from all of the different digests and activation methods. The Coon study used ETD, CAD, and HCD. Maybe this is where the ML assisted informatics increases gains. The sequence coverage issue has direct relevance to ongoing challenges in proteoform characterization and unbiased detection of post-translational modifications, especially in the absence of enrichment.

In summary, the study is well done but overly crammed with data and information and would be better suited to two papers—one on fragmentation of peptides created by these proteases and ion activation methods and one on the informatics.

SPECIFIC COMMENTS:

Introduction focuses on mass spectrometry techniques rather than what has been done with software approaches as if the “slate is blank”. And the inability to do both topics justice in a thorough review of the literature just illustrates that this should be two papers. I realize the Nature Methods format may restrict the introduction but that does a disservice to the relevant prior art that could reduce novelty of the paper and inform readers of prior research. Match assessment and rescoring has been around for at least 25 years as has the idea of fragment ion prediction. As these are scholarly works to move the field forward details will matter. The use of this suite of ion activation methods on non-tryptic peptides would make for an interesting paper unto itself, but a thorough analysis of such data will be lost in this format.

All source code should be available on GitHub. This should be requirement for all academic software that is published in an academic journal. The fundamental requirements for all science are replication and the ability to extension.

Reviewer #2 (Remarks to the Author):

In their new manuscript, N. Levin et al have made claims about improved MS/MS prediction models to be used in conjunction with the Omnitrap's various fragmentation modes. Unfortunately, the text provided far too little information for connecting the three sample types they used to the analysis itself. It is insufficiently clear to me whether training and testing data were kept strictly separate. Their language about improved recovery of true positives is currently misleading. Data Availability is a uncomfortable question mark, at present.

It is worth noting that when the reader reaches the section titled "Large-scale multi-enzyme Omnitrap-LCMS analysis," the reader still has no notion what type of sample the paper will feature. As a result, the reader doesn't know the extent to which offline fractionation or multiple digestion is useful. I am guessing that some type of human cell line is being probed, but that's just my speculation. When I encountered the text "In total, EID data had the least number of PSMs (approximately 900k)," I had no notion how many EID tandem mass spectra were collected, only that twenty pooled fractions per sample (thus twenty RAW files) were used. Were there technical or biological replicates? I am glad that the authors provide the percentages for "efficiency of peptide sequencing," by which they mean the percentage of all MS/MS scans assigned to a peptide sequence, later in the paragraph.

I suggest that the authors report the average and the maximum frequency of MS/MS production by each of the methods. Since I

do not have direct experience with ECD, UVPD, and EID, I don't know their impact on MS/MS production frequency.

I was puzzled by this sentence: "Clearly, increasing the search space by adding more types of fragments results in greatly improved hyperscores for both EID and UVPD." Increasing the number of fragments predicted for a peptide sequence increases the chance of false-positive matching. Generally, our field uses "search space" to communicate the set of peptide mass ladders that get compared to an MS/MS rather than the number of fragment ions typically predicted for those mass ladders; predicting more fragments per peptide doesn't increase the size of this set. MSFragger's ion index approach may be the rationale for which this manuscript describes it as a search space.

I would expect that the high improvement in chymotrypsin data results from the badness of intensity prediction that assumes y ions are twice the intensity of b ions. This is not a bad estimate in peptides that end in Arg or Lys, but it's not a useful estimate in peptides that result from chymotrypsin. Since the intensity model created here is built from chymotryptic peptides, it makes sense that its intensity predictions would be much more applicable.

Did the authors attempt to segregate the experiments upon which their intensity models were trained from the experiments upon which their intensity models were tested? If the same spectra are being used for training as for testing, that's obviously a weakness.

The authors talk about the "recovery of true positive PSMs" with this language: "For that we compared the numbers of TPs identified in each method with the total possible number of TPs across a range of FDR thresholds before and after rescoring." They conclude "The analysis shows that data-driven rescoring using the pan-fragmentation Prosit model recovers essentially all TPs from the initial MSFragger search results." It sounds to me that they are making a claim of near-perfect "recall," but that is quite unlikely. The number of spectra that have been matched to the correct peptide but with too high an estimated q-value is unknown. Instead, I believe the authors are using a much smaller denominator; are they talking about only the PSMs that are retained by any FDR threshold? They need to edit the language in this section to make the language more accurately reflect what they have accomplished.

The last paragraph of the Results is the first place (!) that the authors talk about what kind of data they've used. I see human cell line "Expi293F", E. coli lysates, and A. thaliana seedlings. In Methods, the authors mention that chloracetamide was used to alkylate the human and E. coli sulfhydryls after reduction, but they omit the alkylation information for A. thaliana. Based upon what they've written, I believe that bRPLC (pooling from 80 fractions to 20 samples) was used only on human peptides. Sometimes they employed an Exploris, and sometimes they used an Ascend. Sometimes they used DDA, and sometimes they used DIA. Which of the three species were run on which of the instruments in which data collection strategy? The text is far too jumbled for the reader to discern that. The fact that the authors' first mention of the three species is in the final paragraph of the Results simply baffles me. How is the reader to understand which of these species' proteins are used in which ways above?

I hoped that by looking at the list of RAW files produced for this study I would be able to learn a bit more about how these three species figure into this research. Unfortunately, the authors make no mention of ProteomeXchange or PRIDE or MassIVE. I believe that they haven't made their RAW data available, and that's simply not in keeping with proteomics community standards. Nature Methods requires a Data Availability statement, and the absence of one here is unacceptable.

The authors may want to check this text for accuracy: "ETciD spectra were acquired in the linear ion trap of the Orbitrap Ascend. Products of ETciD were mass-analysed in the Orbitrap at 7500 resolving power." I believe they meant to write that CID was carried out in the linear ion trap, but measurement of the fragments took place in the Orbitrap. Similarly in "HCD spectra were acquired in the HCD cell of the Orbitrap Exploris," I think they intended to say that HCD took place in the HCD cell, but measurement of the fragment ions took place in the Orbitrap. Finally, in "EID, UVPD and ECD DIA spectra were acquired in the Omnitrap" I believe they're talking about dissociation rather than about MS/MS measurement.

The authors report file names for their UniProt databases, but it's unclear whether these represent all proteins matching the species tag in UniProt, in SwissProt only, or perhaps represent reference proteomes. They don't mention the original numbers of protein entries in the FASTA, and they don't mention whether or not they added contaminant proteins. They didn't make the FASTAs available.

Reviewer #3 (Remarks to the Author):

In "Integrating Alternative Fragmentation Techniques into Standard LC-MS Workflows Using a Single Deep Learning Model Enhances Proteome Coverage," the authors present the use of a novel instrument, the "Orbitrap-Omnitrap," applied toward bottom-up proteomics. In the manuscript, the authors characterize the mass spectra acquired for various fragmentation methods and for multiple enzymes. Furthermore, they leverage this data to create a deep learning model to predict the fragment ion intensities of peptides for these alternative fragmentation techniques and enzymes; and they show that this model improves the ability for tools to detect peptides from mass spectra originating from these conditions.

Overall, I found the paper to be well written and interesting. I myself am not an expert on fragmentation methods; hence I restrict my comments to the computational methods employed. Although promising, I think there are methodological questions that must be clarified to assure that the methods presented work as expected. I also have questions about the applicability of the deep learning model beyond the novel instrument used in the manuscript. I would recommend this manuscript be accepted upon major revision.

Major Issues

1. The data splits used for the deep learning model - My understanding is that the training, validation, and test splits for this work were generated at the PSM level, rather than at the peptide sequence level. However, this likely leads to some data leakage in several ways: (1) Multiple spectra for the same peptide may appear in both the training and validation sets with the same fragmentation types, essentially as duplicate examples. (2) If the mass spectra from the same peptide sequence are in the training set and test set for different fragmentation types, similarities between the fragmentation methods may be enough to score well in the test set example. I recommend retraining the model after creating data splits based on peptide sequence rather than PSM. If this is already the case, it should be clarified within the manuscript.
2. Data sources for the deep learning model - Please clarify whether all of the data used for training and evaluating the deep learning model were acquired on the same instrument. If so, such a model likely has limited applicability outside of the specialized instrument used in this study. If only this instrument was used for the training and evaluation, I recommend at minimum demonstrating that it generalizes to another public dataset using one of the alternative fragmentation methods.
3. Application of the deep learning model to rescoring a DDA dataset - In the "Rescoring of the UVPD, ECD and EID data using fragment intensities predictions" section, please clarify whether the dataset that was rescored was the same data dataset that was used to train the model. If they are the same dataset, then I recommend acquiring a new dataset to which to apply rescoring with the deep learning model.
4. Add a negative control to experiments using the new deep learning model - Figure 3C shows the Pearson correlation of the predicted spectra with the acquired spectra. I recommend including a negative control of calculating these correlations using a prior Prosit model's predictions. This should provide a baseline by which to show improvement of the presented model. Furthermore, this baseline would be valuable to add to the experiments in Figure 4 and 5 to demonstrate that the improvement of peptide detection is due to the learned model, not merely switching to Oktoberfest or using a generic predicted library.

Minor Issues

1. Some figures (such as Figure 2D and E) are scatter plots with a high degree of over-plotting. Consider reducing the alpha or adding marginal histograms to help visualize the density of the data.
2. What are the "True Positives" described for Figure 4d? Generally the only true positive examples we have in proteomics are peptides that have been spiked-in.

Version 1:

Decision Letter:

Our ref: NMETH-A61633A

24th Nov 2025

Dear Dr. Mohammed,

Thank you for submitting your revised manuscript "Integrating Alternative Fragmentation Techniques into Standard LC-MS Workflows Using a Single Deep Learning Model Enhances Proteome Coverage" (NMETH-A61633A). It has now been seen by the original referees and their comments are below. The reviewers find that the paper has improved in revision, and therefore we'll be happy in principle to publish it in Nature Methods, pending minor revisions to satisfy the referees' final requests and to comply with our editorial and formatting guidelines.

TRANSPARENT PEER REVIEW

Please note: we allow redactions to authors' rebuttal and reviewer comments in the interest of confidentiality. If you are concerned about the release of confidential data, please let us know specifically what information you would like to have removed. Please note that we cannot incorporate redactions for any other reasons. Reviewer names will be published in the peer review files if the reviewer signed the comments to authors, or if reviewers explicitly agree to release their name. For more information, please refer to our <https://www.nature.com/documents/nr-transparent-peer-review.pdf> target="new">FAQ page.

ORCID

Author names using non-Roman characters

Nature Portfolio journals can support presentation of author names using non-Roman characters in the HTML version of the article. If you wish to, please include author names in parentheses after the Roman-character spelling; [see example online here](https://www.nature.com/articles/s44222-024-00258-2). Currently supported scripts are: Arabic, Chinese, Cyrillic, Devanagari, Greek, Hebrew, Hangul, Japanese and Persian. You will be asked to verify the rendering is correct at proof stage.

Sincerely,
Arunima

Arunima Singh, Ph.D.
Senior Editor
Nature Methods

Reviewer #1 (Remarks to the Author):

The authors have addressed most of my concerns. The other concern was there was too much going in the paper with the topic of fragmentation and software, but the authors made their arguments and I will defer to them. In my experience if a paper has too many major "topics" one of them gets lost. I do like the fact they followed up on the fragmentation issue as one potential advantage to the EAD activation methods is the ability to produce better fragmentation on non-tryptic peptides. They didn't see preference for any particular amino acid which is what I would hope for in an ergodic activation method.

Reviewer #2 (Remarks to the Author):

In revising their manuscript, Levin and coauthors have improved their description of the samples from which their source data have been produced, and I believe the other changes have made their claims of improvement much clearer, as well. For projects requiring alternative proteases like chymotrypsin, I believe this project will have opened a door to better information retrieval. The availability of improved fragmentation models for EID, ECD, and UVPD will also be welcomed by groups with instruments that can produce those.

Given that the number of instruments currently installed in labs that can currently produce ETD or ETCiD data, I would have expected the Ascend ETCiD data to receive a bit more attention (it seems to be absent from all five figures). Instead, the lion's share goes to dissociation modes of the Omnitrap. I think this is a missed opportunity since many labs would be affected today by the ability to maximize information yield from ETD-family dissociation.

This sentence would seem to claim that EID hyperscores are lower than EID hyperscores: "UVPD exhibits marginally higher hyperscores in the low-m/z range than HCD, and EID shows lower hyperscores in the high-m/z range than UVPD and EID."

The authors seem to assert that higher hyperscores are improved hyperscores. A higher score range overall does not mean that discrimination of hyperscores has improved, though.

Shouldn't the labels of radical fragments contain the dot (IUPAC RC-81)?

My eyebrow rose when I encountered the phrase "exceptionally impressive." A term like "notable" would seem a bit more objective.

Reviewer #3 (Remarks to the Author):

The authors have adequately addressed all of my concerns. Thank you.

Version 2:

Decision Letter:

24th Feb 2026

Dear Shabaz,

I am pleased to inform you that your Article, "Integrating Alternative Fragmentation Techniques into Standard LC-MS Workflows Using a Single Deep Learning Model Enhances Proteome Coverage", has now been accepted for publication in Nature Methods. The received and accepted dates will be June 25, 2025 and February 24, 2026. This note is intended to let you know what to expect from us over the next month or so, and to let you know where to address any further questions.

Over the next few weeks, your paper will be copyedited to ensure that it conforms to Nature Methods style. Once your paper is typeset, you will receive an email with a link to choose the appropriate publishing options for your paper and our Author Services team will be in touch regarding any additional information that may be required. It is extremely important that you let us know now whether you will be difficult to contact over the next month. If this is the case, we ask that you send us the contact information (email, phone and fax) of someone who will be able to check the proofs and deal with any last-minute problems.

Authors may need to take specific actions to achieve compliance with funder and institutional open access mandates.

If your research is supported by a funder that requires immediate open access (e.g. according to [Plan S principles](https://www.springernature.com/gp/open-science/plan-s-compliance) or the [NIH public access policy](https://www.springernature.com/gp/open-science/us-federal-agency-compliance)) then you should select the gold OA route, and we will direct you to the compliant route where possible. Because authors warrant under our subscription licensing terms that they haven't committed to licensing any version of their article under a licence inconsistent with the terms of our agreement – including the applicable embargo period – publication under the subscription model isn't suitable for authors whose funders require no embargo.

If you are active on Twitter/X or Bluesky, please e-mail me your and your coauthors' handles so that we may tag you when the paper is published.

Best regards,
Arunima

Arunima Singh, Ph.D.
Senior Editor
Nature Methods

** Visit the Springer Nature Editorial and Publishing website at <http://editorial-jobs.springernature.com?>

utm_source=ejP_NMeth_email&utm_medium=ejP_NMeth_email&utm_campaign=ejp_Nmeth">www.springernature.com/editorial-and-publishing-jobs for more information about our career opportunities. If you have any questions please click here.**

Reviewer #1 (Remarks to the Author):

A dataset was generated using the Omnitrap platform interfaced to an Orbitrap and its diverse ion activation methods. To enhance the study's scope and relevance, a variety of specific proteases were employed to digest proteins. Although chymotrypsin is often regarded as a non-specific protease, Promega has demonstrated—through recent marketing literature—that a highly purified version exhibits substantial specificity. What makes the use of these “alternative” proteases particularly interesting is their relative neglect within the proteomics field. This is largely due to the fact that they do not yield the clean fragmentation patterns typically produced by peptides digested with trypsin. Nevertheless, their strategic utility is undeniable. For example, Josh Coon’s lab published a study in *Nature Biotechnology** showing that the use of multiple proteases enabled high sequence coverage across various cell lines. Even so, the effort plateaued at approximately 80% sequence coverage—highlighting that even combinatorial use of proteases has limitations to achieve 100% sequence coverage. In my view, the field’s reluctance to adopt these proteases stems from the increased complexity they introduce in database searching.

Therefore, a detailed investigation of the fragmentation patterns generated by these proteases under various ion activation conditions would be a valuable contribution to the field. Such a study could illuminate how different fragmentation methods preferentially target specific amino acid binds for fragmentation, especially in the absence of strongly basic residues within a peptide sequence.

Thank you for this suggestion. We agree that potentially our dataset allows such patterns to be discerned. In the original manuscript we focused on the micro (types of fragments) and the macro (general levels of these fragments across entire datasets). Inspired by your suggestion we attempted to extract patterns for each activation method and enzyme combination. Inspecting peptide length distributions (new additional panels in Supplementary Figure 6) we noticed that certain peptide lengths were abundant and common between each sub-dataset. Choosing peptides with 12 AAs, we plotted normalised intensities of fragments for doubly and triply charged peptide precursors against fragmentation position (Supplementary Figures S34, S35). The results allowed clear distinct patterns for each combination. The patterns recreated the well understood CID and ETD spectra of tryptic peptides, but we could see clear trends in all conditions. We also created weblogs of the same peptide populations (Supplementary Figures S36,S37), unfortunately there we saw little preference for certain amino acids which suggested that a well-tuned identification progress minimises sequence biases. These additional figures have been complemented with an extended discussion in Supplementary Notes - “Enzyme-specific fragmentation patterns”.

This could form the basis of a compelling manuscript, provided sufficient detail is included to illustrate how fragmentation characteristics correlate with peptide composition. A separate paper focused on improved methods for identifying peptides generated by these proteases—along with a thorough description of the computational approach—would also be of substantial value to the community. Combined, these two interesting topics are diluted.

We appreciate the reviewer’s excitement for fragmentation characteristics that our data affords. As mentioned above, we have taken a deeper dive into this topic and provided an extensive Supplementary Note and Figures. We hope this structure allows the manuscript flow to be maintained while allowing

those enthusiastic for fragmentation readers to enjoy a more thorough analysis in the supplementary section.

The study is well done, and the paper is clear and well written. They report a 10% improvement in peptide identifications which is not much of an advance—it may be better characterized as experimental variation.

Thank you for the warm welcome of our manuscript. We appreciate that a 10% improvement out of context does appear modest. To be accurate the generated analyses require describing contexts for every activation technique and enzyme choice which would require quite a considerable amount of text. We opted to generalise to reduce perceived repetition and, furthermore, we opted to be conservative with reporting numbers. We'd like to point out though, that the improvements in identification vary depending on the enzyme and dissociation approach and can be as high as 40.5% on the PSM level and 35.9% on the peptide level. Furthermore, considering the size of the dataset, a 10% gain corresponds to tens of thousands of additional PSMs which is a significant quantity of data.

A more insightful metric might be the gain in sequence coverage, as demonstrated by the Coon study by combining the results from all of the different digests and activation methods. The Coon study used ETD, CAD, and HCD. Maybe this is where the ML assisted informatics increases gains. The sequence coverage issue has direct relevance to ongoing challenges in proteoform characterization and unbiased detection of post-translational modifications, especially in the absence of enrichment.

We were also inspired by the Coon group's Nature Biotechnology paper. In Supplementary Figures S39-S43, we discuss how each condition independently and collectively contributes to sequence coverage. In an extended discussion we point that we also observe increased coverage of proteome as we combine conditions. However, our improvements are far more modest when compared to the Coon work. Our use of a single cell line leaves us with a dynamic range issue causing 'low abundance proteins' to be poorly covered by all conditions. Coon and co-workers alleviated the issue through the use of a strategic panel of cell lines and deeper fractionation, so we are not able to replicate the dramatic sequence coverage complementary of enzymes across an entire proteome BUT we do show, as expected, the enzymes are complementary.

“The rescoring data provided an opportunity to inspect the efficacy of each enzyme and dissociation technique for proteome analysis (Supplementary Notes, Supplementary Fig. S31-S37). Trypsin, as expected, identified the most PSMs, peptides, and proteins for every fragmentation technique. Chymotrypsin demonstrated the next best result, with LysC and LysN a little further behind, replicating previous trends observed for CID and ETciD data (Supplementary Fig. S31a, S32a).⁴⁴⁻⁴⁶ The enzyme GluC clusters with LysN, appearing to be slightly superior or inferior depending on the dissociation technique. Average protein sequence coverage was similar for each fragmentation technique (Supplementary Fig S33). In order to assess complementarity at the protein sequence level we represented our data at the amino acid level. In general terms, when comparing the complementarity of trypsin against its alternatives, we saw substantial improvements in proteome coverage for all fragmentation techniques (Supplementary Fig. S31b, S32b); in fact, the unique combined coverage for LysN, LysC, GluC, and chymotrypsin was more than that for trypsin. These observations echo previous work demonstrating the complementarity of enzymes for improving sequence coverage.^{39,44-46} It should be noted that each trypsin fraction was essentially analysed by LCMS four times, and a more exhaustive LCMS analysis would not significantly increase proteome coverage, and so the amount of

analysis time between the other enzymes versus trypsin is not an important factor in the comparison. Further analysis of unique coverage for each fragmentation technique showed that UVPD produced the most amount of unique data, with HCD and ECD close behind, and EID the least (Supplementary Fig. S31c). However, UVPD had significant overlap with EID which might be a reason for weak unique proteome coverage result for EID (Supplementary Figure S31c)."

In summary, the study is well done but overly crammed with data and information and would be better suited to two papers- one on fragmentation of peptides created by these proteases and ion activation methods and one on the informatics.

Thank you for the high opinion of our work. We appreciate the fact that the information density in the manuscript may appear unwieldy, but we believe that it is these data together with the deep learning model that complement each other and reflect the goal of the project, which is to create a robust proteomics framework for analysis of non-CID data (made possible by the Omnitrap instrument).

SPECIFIC COMMENTS:

Introduction focuses on mass spectrometry techniques rather than what has been done with software approaches as if the "slate is blank". And the inability to do both topics justice in a thorough review of the literature just illustrates that this should be two papers. I realize the Nature Methods format may restrict the introduction but that does a disservice to the relevant prior art that could reduce novelty of the paper and inform readers of prior research. Match assessment and rescoring has been around for at least 25 years as has the idea of fragment ion prediction. As these are scholarly works to move the field forward details will matter.

As the reviewer notes we were constrained by word count for our introduction; however, we agree with the reviewer that we need a more balanced summary of past work with respect to data analysis. We have now extended the introduction to reflect a bit better the prior work on data analysis software:

"In recent years, there has been significant progress in the development of more accurate tools for analysis of bottom-up proteomics data. In particular, it has been shown that using data-driven rescoring pipelines based on predictions of peptide properties such as their retention time or fragmentation spectra greatly increase numbers of identifications compared to standard database searches. Among deep learning-based tools, Prosit and pDeep have gained the highest popularity for improving peptide identification through fragmentation spectrum prediction. The vast majority of these predictors are restricted to collisional data, while ExD methods and UVPD remain largely unsupported. To date, only one model, PredFull, has demonstrated the ability to predict alternative ion types from ETD data; nevertheless, it still lacks sufficient training to robustly support UVPD and other ExD fragmentation methods such as ECD and EID."

The use of this suite of ion activation methods on non-tryptic peptides would make for an interesting paper unto itself, but a thorough analysis of such data will be lost in this format.

As mentioned above we have now incorporated an extended discussion on the topic.

All source code should be available on GitHub. This should be requirement for all academic software that is published in an academic journal. The fundamental requirements for all science are replication and the ability to extension.

We wholeheartedly agree with the reviewer on providing the underlying data and code for published work. Unfortunately, we initially provided the required information for access in the letter to the editor and not the manuscript. We did correct this mistake by providing an updated manuscript now containing the relevant information but unfortunately there was a mix-up in the editorial office. This issue has been corrected for the resubmission. The below information can now be found in the manuscript.

“Mass spectrometry proteomics data have been deposited to the ProteomeXchange Consortium via the PRIDE partner repository with the dataset identifiers PXD065289. Project accession: PXD065289 Username: reviewer_pxd065289@ebi.ac.uk Password: FfHhPs2gtZ6p

Source code and scripts are available on GitHub: Oktoberfest (<https://github.com/wilhelm-lab/oktoberfest>), Koina (<https://github.com/wilhelm-lab/koina>), MSBooster (<https://github.com/Nesvilab/MSBooster>) and the source code for training the model (https://github.com/wilhelm-lab/Prosit_multifrag).”

Reviewer #2 (Remarks to the Author):

In their new manuscript, N. Levin et al have made claims about improved MS/MS prediction models to be used in conjunction with the Omnitrap's various fragmentation modes. Unfortunately, the text provided far too little information for connecting the three sample types they used to the analysis itself.

It is insufficiently clear to me whether training and testing data were kept strictly separate.

Their language about improved recovery of true positives is currently misleading.

Thank you for catching this, we have adjusted the corresponding sections to improve clarity. We also clarify that we are referring to estimated true positives, as the actual true positives are almost never known. These estimates rely on the assumption of the target–decoy approach, where the number of decoys is expected to approximate the number of false positive targets, thus, the number of estimated true positives is the number of targets minus the number of decoys.

“To explore the reasons of the varying numbers of gains observed, we investigated the recovery of estimated true positive (TP) PSMs. For that, we compared the number of estimated TPs across a range of FDR thresholds (by subtracting the number of decoy PSMs from the number of target PSMs at different FDR cutoffs) before and after rescoring with the total number of estimated TPs in the dataset that could be recovered from the initial search results by subtracting the total number of decoys from the total number of target PSMs (Fig. 4d, Supplementary Fig. S30). At 1% PSM-level FDR, rescored ECD, EID, and UVPD recovered more than 97% of possible TPs, while the original database searches extracted approximately 95% in ECD, 87% in EID, 85% in UVPD, and 84% in HCD. At a stricter FDRs of 0.01%, the results after rescoring still captured over 75% of all estimated possible TPs, with ECD showing the highest proportion approaching 85%. At the same FDR level, initial base searches identified less than 70% of possible TPs in ECD and less than 55% in all other dissociation methods (Fig. 4d). The analysis shows

that data-driven rescoring using the pan-fragmentation Prosit model substantially increases the proportion of estimated TP PSMs retained at stringent thresholds, approaching saturation of the set of PSMs recoverable from the initial MSFragger search results. It is important to note that further correct identifications, e.g. from modified peptides not considered in the initial search, cannot be considered in the estimation of the number of TPs."

Data Availability is an uncomfortable question mark, at present.

As pointed out above, there was mix-up with the manuscript submission and dissemination. All data is available in the public realm, information above.

It is worth noting that when the reader reaches the section titled "Large-scale multi-enzyme Omnitrap-LCMS analysis," the reader still has no notion what type of sample the paper will feature. As a result, the reader doesn't know the extent to which offline fractionation or multiple digestion is useful. I am guessing that some type of human cell line is being probed, but that's just my speculation.

We have now clarified our experimental set up in several locations in the text and figures.

We inserted the following text on page 3 of the manuscript (in the "Development of Omnitrap UVPD, ECD and EID LCMS methods" section): "Unless otherwise specified, human Expi293F cells were used as analyte throughout the paper."

When I encountered the text "In total, EID data had the least number of PSMs (approximately 900k)," I had no notion how many EID tandem mass spectra were collected, only that twenty pooled fractions per sample (thus twenty RAW files) were used. Were there technical or biological replicates?

Each fraction or sample was analysed once in each condition. The goal of the data was to generate diversity and we would measure reproducibility through the test fraction of the datasets.

In addition, we adjusted the text on page 4 regarding the size of the dataset which now reads:

"In total, each fragmentation technique produced between approximately 3.5 million MS2 spectra in ECD and 4.5 million MS2 spectra in HCD across 5 enzymes, 20 fractions per enzyme (Fig. 2c). EID data had the least number of PSMs (approximately 900k), while UVPD, which has the fastest acquisition rate among all Omnitrap techniques studied here (~6.3 MS2 scans per second on average), had 1,141k (Figure 2c). Surprisingly, the charge-dependent ECD came closest to UVPD with 1,070k PSMs, even though its scan rate (~5.2 MS2 spectra per second) was essentially the same as in EID. HCD showed the highest numbers with 1,160k PSMs acquired using 60-min gradients at the rate of on average ~13 MS2 scans per second. Pleasingly, the efficiency of peptide sequencing by EID (24.8%) and UVPD (25.6%), expressed as the ratio between the numbers of confidently identified PSMs and acquired MS2 scans, is essentially the same as by HCD (24.9%) while the efficiency of sequencing by ECD (30.3%) was the best (Fig. 2c). This was surprising considering the relative inefficiency of ECD for doubly charged peptides which represent a substantial subset of identified peptides (Supplementary Fig. S6)."

I am glad that the authors provide the percentages for "efficiency of peptide sequencing," by which they mean the percentage of all MS/MS scans assigned to a peptide sequence, later in the paragraph. I suggest that the authors report the average and the maximum frequency of MS/MS production by each of the methods. Since I do not have direct experience with ECD, UVPD, and EID, I don't know their impact on MS/MS production frequency.

We have carried out the suggested analysis, the result and adjusted text can be found above.

I was puzzled by this sentence: "Clearly, increasing the search space by adding more types of fragments results in greatly improved hyperscores for both EID and UVPD." Increasing the number of fragments predicted for a peptide sequence increases the chance of false-positive matching. Generally, our field uses "search space" to communicate the set of peptide mass ladders that get compared to an MS/MS rather than the number of fragment ions typically predicted for those mass ladders; predicting more fragments per peptide doesn't increase the size of this set. MSFragger's ion index approach may be the rationale for which this manuscript describes it as a search space.

We agree with the reviewer's point that we have misused the term. We have adjusted the wording as follows:

"Clearly, adding more types of fragments results in greatly improved hyperscores for both EID and UVPD, indicating larger numbers of dissociated bonds and data rich spectra."

I would expect that the high improvement in chymotrypsin data results from the badness of intensity prediction that assumes y ions are twice the intensity of b ions. This is not a bad estimate in peptides that end in Arg or Lys, but it's not a useful estimate in peptides that result from chymotrypsin. Since the intensity model created here is built from chymotryptic peptides, it makes sense that its intensity predictions would be much more applicable.

We agree we expected the dramatically different sequence properties of chymotryptic peptides to be better reflected by our model. We, following the suggestion of the reviewers, have performed a deeper dive into the patterns present in spectra. Results are described above.

Did the authors attempt to segregate the experiments upon which their intensity models were trained from the experiments upon which their intensity models were tested? If the same spectra are being used for training as for testing, that's obviously a weakness.

We did not initially for these experiments, as we did not believe that the model's performance would be all that sensitive to the specific experiment in which the spectra were acquired. To test this, we trained two further models where the spectra of one specific enzyme are withheld in the training set and then evaluated on in the test set. This was initially performed for first Chymotrypsin and then LysN.

Withheld enzyme	Median PCC on non-withheld enzyme	Median PCC on withheld enzyme
Chymotrypsin	0.909	0.891
LysN	0.915	0.898

The spectra from the enzyme which the model didn't see during training has a consistent deficit of ~1.8 percentage points. The small disparity indicates that the model has captured the fundamental patterns of the spectra and is not simply reproducing the peculiarities of a given experiment. This bolsters our argument that the model should be applicable for rescoring data originating outside of the experiments

in this study. Please note we were careful with what data was used for training and what for testing. A detailed answer about our processes can be found below.

The authors talk about the "recovery of true positive PSMs" with this language: "For that we compared the numbers of TPs identified in each method with the total possible number of TPs across a range of FDR thresholds before and after rescoring." They conclude "The analysis shows that data-driven rescoring using the pan-fragmentation Prosit model recovers essentially all TPs from the initial MSFragger search results." It sounds to me that they are making a claim of near-perfect "recall," but that is quite unlikely. The number of spectra that have been matched to the correct peptide but with too high an estimated q-value is unknown. Instead, I believe the authors are using a much smaller denominator; are they talking about only the PSMs that are retained by any FDR threshold? They need to edit the language in this section to make the language more accurately reflect what they have accomplished.

We addressed this remark in one of the previous replies.

The last paragraph of the Results is the first place (!) that the authors talk about what kind of data they've used. I see human cell line "Expi293F", E. coli lysates, and A. thaliana seedlings. In Methods, the authors mention that chloroacetamide was used to alkylate the human and E. coli sulfhydryls after reduction, but they omit the alkylation information for A. thaliana. Based upon what they've written, I believe that bRPLC (pooling from 80 fractions to 20 samples) was used only on human peptides.

We greatly appreciate the reviewer's comments regarding our Methods section, and we agree that it indeed was messy and contained errors. We have made the following adjustments to address the reviewer's concerns:

We inserted the following text on page 3 of the manuscript (in the "Development of Omnitrap UVPD, ECD and EID LCMS methods" section): "Unless otherwise specified, human Expi293F cells were used as analyte throughout the paper."

We adjusted the Methods section as follows:

"Protein extraction

Ground Arabidopsis thaliana seedlings were homogenised in 100 mM Tris buffer pH 7.6 containing 4% SDS, 1x Protease inhibitor cocktail (Roche), 1x PhosStop (Roche), 10 mM TCEP, 50 mM CAA, 20 mg/ml PVPP beads (Alfa Aesar) in an ice-cold ultrasonic water bath for 30 min. The homogenised extract was then incubated for 2h in an orbital shaker (400 rpm) at room temperature and clarified by two 10-min room-temperature centrifugation steps at 17,000 g."

The following line was added to the "Proteolysis" subsection:

"Proteins from Expi293F were subjected to proteolysis using either trypsin, chymotrypsin, LysC, LysN or GluC, whereas proteins from A. Thaliana and E.coli were digested only by trypsin..."

We also added a line in the last paragraph of Results ("Application of DIA with all activation techniques and Pan-activation Prosit Model"):

"We carried out LCMS analyses on unfractionated tryptic cell lysate digests from Homo sapiens (Expi293F), Arabidopsis thaliana, and Escherichia coli cells."

Sometimes they employed an Exploris, and sometimes they used an Ascend. Sometimes they used DDA, and sometimes they used DIA. Which of the three species were run on which of the instruments in which data collection strategy? The text is far too jumbled for the reader to discern that. The fact that the authors' first mention of the three species is in the final paragraph of the Results simply baffles me. How is the reader to understand which of these species' proteins are used in which ways above?

We completely rewrote the “LCMS analysis” subsection in Methods, which now reads:

“LCMS DDA

LC-MS/MS data were acquired using an UltiMate 3000 nanoUHPLC system (Thermo Fisher Scientific) coupled either to an Orbitrap Exploris (Thermo Fisher Scientific) equipped with an Omnitrap (Fasmatech) for UVPD, EID, ECD and HCD analyses or to an Orbitrap Ascend Tribrid (Thermo Fisher Scientific) for ETciD analyses. The peptides were trapped on a C18 PepMap100 pre-column (300 µm i.d. x 5 mm, 100 Å, Thermo Fisher Scientific) using solvent A (0.1% formic acid in water), then separated on an in-house packed analytical column (50 µm i.d. x 50 cm in-house packed with ReproSil Gold 120 C18, 1.9 µm, Dr. Maisch GmbH). The composition of solvent B (0.1% formic acid in acetonitrile) changed from 10% to 33% over 30 or 60 min for parameters optimisation experiments and for HCD analyses or from 8% to 28% over 120 min for UVPD, EID and ECD analyses. Full scan MS1 spectra were acquired in the Orbitrap (scan range 400-1300 m/z, resolution 60000, AGC target 300%). Top 20 (40 in HCD) most abundant peptides were selected each round of DDA for fragmentation. EID, UVPD and ECD were performed in the Omnitrap. Their products were mass-analysed in the Orbitrap at 45000 resolving power, the AGC value was set to 200%, and the maximum injection time was set to 64 ms. ETciD was performed in the linear ion trap of the Orbitrap Ascend. Precursor ions were subject to ETD for the following reaction times: 2+ and 3+ for 50 ms, 4+ for 25 ms, 5+ to 7+ for 16 ms. charge-reduced species were further activated using ion-trap CID at 35% of normalised collision energy. Products of ETciD were mass-analysed in the Orbitrap at 7500 resolving power, the AGC value was set to 200%, and the maximum injection time was set to 64 ms. HCD was performed in the HCD cell of the Orbitrap Exploris. Products of HCD were mass-analysed in the Orbitrap at 7500 resolving power, the AGC value was set to 40%, and the maximum injection time was set to 64 ms.

LCMS DIA

Data independent analyses were conducted in the same way as the data dependent analyses, with the following adjustments: The composition of solvent B (0.1% formic acid in acetonitrile) changed from 8% to 20% over 240 min for UVPD, EID and ECD DIA, or from 8% to 28% over 120 for HCD DIA at a flow rate of 100 nL/min. DIA scans were acquired using 4- or 8-m/z isolation windows in the 400-700 m/z range over 0-80 minutes, 500-700 m/z range over 80-160 minutes, and 600-900 m/z range over 160-240 min of a 240-min LC gradient. Products of ECD, EID, HCD and UVPD were mass-analysed in the Orbitrap at 60000 resolving power, the AGC value was set to 2000%, and the maximum injection time was set to 50 ms.”

I hoped that by looking at the list of RAW files produced for this study I would be able to learn a bit more about how these three species figure into this research. Unfortunately, the authors make no mention of ProteomeXchange or PRIDE or MassIVE. I believe that they haven't made their RAW data available, and that's simply not in keeping with proteomics community standards. Nature Methods requires a Data Availability statement, and the absence of one here is unacceptable.

We apologize for the inconvenience. As stated above there was a mix-up with the data dissemination process. In the meantime, we have now also reorganized the PRIDE repository to enhance clarity and accessibility.

The authors may want to check this text for accuracy: "ETciD spectra were acquired in the linear ion trap of the Orbitrap Ascend. Products of ETciD were mass-analysed in the Orbitrap at 7500 resolving power." I believe they meant to write that CID was carried out in the linear ion trap, but measurement of the fragments took place in the Orbitrap.

Mistake has been corrected, see above.

Similarly in "HCD spectra were acquired in the HCD cell of the Orbitrap Exploris," I think they intended to say that HCD took place in the HCD cell, but measurement of the fragment ions took place in the Orbitrap. Finally, in "EID, UVPD and ECD DIA spectra were acquired in the Omnitrap" I believe they're talking about dissociation rather than about MS/MS measurement.

Mistakes have been corrected, see above.

The authors report file names for their UniProt databases, but it's unclear whether these represent all proteins matching the species tag in UniProt, in SwissProt only, or perhaps represent reference proteomes. They don't mention the original numbers of protein entries in the FASTA, and they don't mention whether or not they added contaminant proteins. They didn't make the FASTAs available.

We thank the reviewer for highlighting the need for clarity regarding FASTA. The following lines were added in the corresponding section:

“For Expi293F (Proteome ID: UP000005640), FASTA contains one protein sequence per gene and contains a total of 20,702 protein entries (Swis-Prot 20,390 and TrEMBL 312 protein sequences). E.coli (Proteome ID: UP000000625) FASTA contains a total of 4,489 protein entries (Swis-Prot 4,474 and TrEMBL 15 protein sequences), and A.thaliana (Proteome ID: UP000006548) FASTA contains a total of 39,443 protein entries (Swis-Prot 16,278 and TrEMBL 23,165 protein sequences).”

Reviewer #3 (Remarks to the Author):

In "Integrating Alternative Fragmentation Techniques into Standard LC-MS Workflows Using a Single Deep Learning Model Enhances Proteome Coverage," the authors present the use of a novel instrument, the "Orbitrap-Omnitrap," applied toward bottom-up proteomics. In the manuscript, the authors characterize the mass spectra acquired for various fragmentation methods and for multiple enzymes. Furthermore, they leverage this data to create a deep learning model to predict the fragment ion intensities of peptides for these alternative fragmentation techniques and enzymes; and they show that this model improves the ability for tools to detect peptides from mass spectra originating from these conditions.

Overall, I found the paper to be well written and interesting. I myself am not an expert on fragmentation methods; hence I restrict my comments to the computational methods employed. Although promising, I think there are methodological questions that must be clarified to assure that the methods presented work as expected. I also have questions about the applicability of the deep learning model beyond the novel instrument used in the manuscript. I would recommend this manuscript be accepted upon major revision.

Thank you for your kind comments. We understand that the use of a novel instrument can generate the impression that our observations might possess a uniqueness that does not translate to a general picture or model. Thankfully this is not the case. Our approach and implementation of the activation approaches was to ascertain the fundamental mechanisms and parameters underpinning the fragmentation. Our results can explain ExD and UVPD implementations elsewhere, as we (and others) have already demonstrated for top-down proteomics. There are, of course, second and third order effects associated with each instrument implementation and acquisition choices. This would not be dissimilar to the CID where the underlying mechanisms can be described by the mobile proton model which can provide an excellent first approximation of peptide fragmentation spectra. The instrument energy choices and mass transmission profile will dictate the exact spectrum. These parameters have been identified for CID and have been used to incorporate many CID implementations into currently available models. In the first part of the results section we describe which parameters have effects on UVPD and ExD spectra, which will help parameterise and then incorporate new implementations of these activation techniques, this will necessarily have to be a community wide effort.

Major Issues

1. The data splits used for the deep learning model - My understanding is that the training, validation, and test splits for this work were generated at the PSM level, rather than at the peptide sequence level. However, this likely leads to some data leakage in several ways: (1) Multiple spectra for the same peptide may appear in both the training and validation sets with the same fragmentation types, essentially as duplicate examples. (2) If the mass spectra from the same peptide sequence are in the training set and test set for different fragmentation types, similarities between the fragmentation methods may be enough to score well in the test set example. I recommend retraining the model after creating data splits based on peptide sequence rather than PSM. If this is already the case, it should be clarified within the manuscript.

Prior to splitting the data for train/val/test, the PSMs were de-duplicated such that every spectrum is a unique combination of modified sequence/charge/fragmentation method. These 3 de-duplication criteria were selected on the philosophy that together they comprise a unique input to the model, and thus will produce a unique output. Yes, the same peptide may show up in training and validation, but necessarily with different charge states and/or fragmentation methods. The train/val/test sets are then divided as a random 80/10/10 split. Nevertheless, your point is well taken since peptide sequence is the major input determining the model's prediction, and therefore we must communicate the model's performance for seen and unseen sequences.

To address the model's performance on unseen and seen sequences, we split the model's test set into sequences that were in the training set and those that did not occur in the training set.

	Median in-train sequences PCC Random split	Median out-of-train sequences PCC Random split
HCD	0.946	0.917
UVPD	0.927	0.882
EID	0.888	0.789
ECD	0.917	0.868
Overall	0.913	0.875

Overall the disparity in performance is ~4% and varies by fragmentation method. EID has a nearly 10% deficit with unseen sequences while HCD has a more modest 3% deficit. A corresponding figure in the supplementary materials is provided to show the overall distributions of the in-train and out-of-train sequences in our test set.

Please see Supplementary Notes – “*Evaluation of train/test split*”

2. Data sources for the deep learning model - Please clarify whether all of the data used for training and evaluating the deep learning model were acquired on the same instrument. If so, such a model likely has limited applicability outside of the specialized instrument used in this study. If only this instrument was used for the training and evaluation, I recommend at minimum demonstrating that it generalizes to another public dataset using one of the alternative fragmentation methods.

As stated earlier our intention was to understand and demonstrate the value of alternative fragmentation approaches in bottom-up proteomics. Although our ECD, EID and UVPD datasets are all acquired on one bespoke platform, as stated above, our implementation allows general observations to be ascertained. It is worth stating further, our datasets are a first of kind, only made possible by the construction of the instrument. Our observations and prediction model form part of our argument for advocating more use of these approaches. Nevertheless, this is a very good point. Unfortunately, there aren't any viable ExD and UVPD datasets out there, one of the logical conclusions of our arguments. We understand the desire to demonstrate generality and so we tried a slightly different approach. Electron capture dissociation was initially discovered by Zubarev and McLafferty in the late 90s and was shown

to be an excellent fragmentation technique for proteins and heralded top down proteomics. The use of electrons made it difficult to implement on traditional proteomics instruments. Syka, Coon and Hunt developed electron transfer dissociation (ETD), an approach that allowed implementation on regular 'trapping' instruments and generated similar results to ECD. A large body of work has demonstrated that the two techniques employ the same underlying mechanisms and produce similar data when parameters are appropriately controlled. Utilising the phenomenon that ETD and ECD are closely related we chose this relationship to demonstrate our observations and fundamental predictions are sound. We had generated a similar dataset for ETD as for the other techniques, but we did not focus on it due to the technique already enjoying significant characterisation. We have now made the ETD data more prominent.

Please see Supplementary Note "*Acquisition and Peculiarities of the ETciD data*", "*Enzyme-specific fragmentation patterns*" and Supplementary Figs 17, 25, 28, 32, 34-37

We then subjected the ETD data to the spectral pattern analysis (as described above). We found that the ETD data appeared to be very similar to ECD data corroborating that the two techniques are mechanistically similar (use of electrons to initiate fragmentation through radicals). Indicating that not only are ECD and ETD close but that our model will be able to rationalise other implementations of both ETD and ECD. It is noteworthy this similarity is observed despite the data being generated with quite different approaches and acquired on completely different platforms (ETD in the Thermo Linear Ion Trap vs ECD in the Omnitrap – both acquire spectra in Orbitraps). In other words, the fundamental mechanisms dominate over the implementation and rescoring using our model would lead to improved identification rates. We decided to go one step further and analysed a public ETD dataset. We chose the recent Nature Biotechnology paper by Coon and co-workers with the title "Global detection of human variants and isoforms by deep proteome sequencing". Unfortunately, this public dataset (and pretty much all ETD datasets) acquired the spectra in the linear ion trap (low resolution, low mass accuracy) creating difficulty in performing a robust comparison. A secondary issue was that the authors chose to maximise the complementarity of HCD, ETHcD, and ETD by choosing ETHcd and ETD to focus on the 200-800 m/z range and for ETD to be applied to 3+ and higher charge states. The authors set the low mass cut off for the ETD data to 120, far lower than in our acquisitions which is set by the instrument (at higher values) due our focus on the typical mass range 300-1500 m/z. Still this dataset is the most comprehensive dataset available and acquired on a modern Tribrid instrument. Pleasingly, our analyses (Supplementary Figs S39 to S43) show that our model does possess predictive powers. Our model predicts mostly the same fragments but there is clearly a mass dependent intensity profile shift to lower m/z. The profile shift can be explained by the author acquisition choices. Furthermore, rescoring this dataset led to a modest increase in identifications indicating the utility of the current model. Our model possesses high predictive power but will require augmenting with datasets acquired with a range of acquisition parameters (that we have identified and described) to be better applicable to the wide range of instruments available. This would be no different to what has happened for CID but would require a community wide effort as more vendors release instruments with these activation methods.

Please see Supplementary Notes "*Applicability of the deep learning model to other datasets*"

3. Application of the deep learning model to rescoring a DDA dataset - In the "Rescoring of the UVPD,

ECD and EID data using fragment intensities predictions" section, please clarify whether the dataset that was rescored was the same data dataset that was used to train the model. If they are the same dataset, then I recommend acquiring a new dataset to which to apply rescoring with the deep learning model.

The initial identification dataset used for training is part of the total DDA set that was rescored. The new identifications gained by rescoring were not in the initial training dataset and were subjected to a target-decoy analysis. The comprehensiveness of our dataset would have required quite a different sample set for removing any concerns. We chose not only to use a different sample set but also samples from different species (*Arabidopsis Thaliana* and *Escherichia coli*) alongside a human sample which were then subjected to DIA. We opted to use DIA umpire for analysis which converts the data to a more DDA-like analysis. Pleasingly, we saw similar improvements for *A. Th.* to the human samples.

4. Add a negative control to experiments using the new deep learning model - Figure 3C shows the Pearson correlation of the predicted spectra with the acquired spectra. I recommend including a negative control of calculating these correlations using a prior Prosit model's predictions. This should provide a baseline by which to show improvement of the presented model. Furthermore, this baseline would be valuable to add to the experiments in Figure 4 and 5 to demonstrate that the improvement of peptide detection is due to the learned model, not merely switching to Oktoberfest or using a generic predicted library.

We thank the reviewer for the points on the negative control. We evaluated the Prosit 2020 HCD intensity prediction model on the same datasets used for rescoring and compared Pearson correlation with our multi-fragmentation model. As shown in Supplementary Figure S19, the Prosit 2020 HCD model yields substantially lower correlations. For ECD, the Prosit 2020 HCD showed a broad distribution, and this can be explained by the fact that the HCD model fails in capturing the c,z ion for ECD. In contrast, for EID and UVPD, which generate b/y ions, retain partial predictive capability. This explains why the correlation distributions for EID and UVPD are higher and narrower compared to ECD, though still consistently outperformed by the Prosit 2025 multi-fragmentation model.

Please see Supplementary Note "*Negative control test of the MultFrag deep learning model*"

Minor Issues

1. Some figures (such as Figure 2D and E) are scatter plots with a high degree of over-plotting. Consider reducing the alpha or adding marginal histograms to help visualize the density of the data.

We opted for replacing old scatter plots with contour plot diagrams that show the same data. We also added three Supplementary Figures (S8, S9, S10) which show the same data broken down by charge state. To reflect these changes, we slightly amended the text (last paragraph in the "*Large-scale multi-enzyme Omnitrap-LCMS analysis*" section):

2. What are the "True Positives" described for Figure 4d? Generally the only true positive examples we have in proteomics are peptides that have been spiked-in.

We addressed this remark in one of the previous replies. (See Reviewer 2' point 1)

Reviewer #1:

Remarks to the Author:

The authors have addressed most of my concerns. The other concern was there was too much going in the paper with the topic of fragmentation and software, but the authors made their arguments and I will defer to them. In my experience if a paper has too many major "topics" one of them gets lost. I do like the fact they followed up on the fragmentation issue as one potential advantage to the EAD activation methods is the ability to produce better fragmentation on non-tryptic peptides. They didn't see preference for any particular amino acid which is what I would hope for in an ergodic activation method.

Thank you and acknowledged.

Reviewer #2:

Remarks to the Author:

In revising their manuscript, Levin and coauthors have improved their description of the samples from which their source data have been produced, and I believe the other changes have made their claims of improvement much clearer, as well. For projects requiring alternative proteases like chymotrypsin, I believe this project will have opened a door to better information retrieval. The availability of improved fragmentation models for EID, ECD, and UVPD will also be welcomed by groups with instruments that can produce those.

Given that the number of instruments currently installed in labs that can currently produce ETD or ETCiD data, I would have expected the Ascend ETCiD data to receive a bit more attention (it seems to be absent from all five figures). Instead, the lion's share goes to dissociation modes of the Omnitrap. I think this is a missed opportunity since many labs would be affected today by the ability to maximize information yield from ETD-family dissociation.

This sentence would seem to claim that EID hyperscores are lower than EID hyperscores: "UVPD exhibits marginally higher hyperscores in the low-m/z range than HCD, and EID shows lower hyperscores in the high-m/z range than UVPD and EID."

Corrected – "EID shows lower hyperscores in the high-m/z range than UVPD and HCD."

The authors seem to assert that higher hyperscores are improved hyperscores. A higher score range overall does not mean that discrimination of hyperscores has improved, though.

We are simply using hyperscores to compare spectral information not confidence or discrimination. The comparisons are to allow the reader to gain understanding where are the gains and losses of information for each technique.

Shouldn't the labels of radical fragments contain the dot (IUPAC RC-81)?

We considered what is the clearest way to annotate fragments. Unfortunately, communities for each activation technique has bespoke nomenclature to make it easier to annotate for their own technique which causes confusion when discussing more than one technique. We chose to use a recently

recommended nomenclature outlined in supplementary table 1 which has been designed to be universal.

My eyebrow rose when I encountered the phrase "exceptionally impressive." A term like "notable" would seem a bit more objective.

Texted changed to 'notable':

Reviewer #3:

Remarks to the Author:

The authors have adequately addressed all of my concerns. Thank you.

Thank you.